# Interdependence of Kinetics and Fluid Dynamics in the Design of Photocatalytic Membrane Reactors

**DOI:** 10.3390/membranes12080745

**Published:** 2022-07-29

**Authors:** Vimbainashe Chakachaka, Charmaine Tshangana, Oranso Mahlangu, Bhekie Mamba, Adolph Muleja

**Affiliations:** Institute for Nanotechnology and Water Sustainability, College of Science, Engineering and Technology, University of South Africa, Florida, Johannesburg 1709, South Africa; 14794756@mylife.unisa.ac.za (V.C.); tshansc@unisa.ac.za (C.T.); mahlaot@unisa.ac.za (O.M.); mambabb@unisa.ac.za (B.M.)

**Keywords:** mathematical model, geometry, optical fiber, energy efficiency, process intensification, water treatment, upscaling

## Abstract

Photocatalytic membrane reactors (PMRs) are a promising technology for wastewater reclamation. The principles of PMRs are based on photocatalytic degradation and membrane rejection, the different processes occurring simultaneously. Coupled photocatalysis and membrane filtration has made PMRs suitable for application in the removal of emerging contaminants (ECs), such as diclofenac, carbamazepine, ibuprofen, lincomycin, diphenhydramine, rhodamine, and tamoxifen, from wastewater, while reducing the likelihood of byproducts being present in the permeate stream. The viability of PMRs depends on the hypotheses used during design and the kinetic properties of the systems. The choice of design models and the assumptions made in their application can have an impact on reactor design outcomes. A design’s resilience is due to the development of a mathematical model that links material and mass balances to various sub-models, including the fluid dynamic model, the radiation emission model, the radiation absorption model, and the kinetic model. Hence, this review addresses the discrepancies with traditional kinetic models, fluid flow dynamics, and radiation emission and absorption, all of which have an impact on upscaling and reactor design. Computational and analytical descriptions of how to develop a PMR system with high throughput, performance, and energy efficiency are provided. The potential solutions are classified according to the catalyst, fluid dynamics, thickness, geometry, and light source used. Two main PMR types are comprehensively described, and a discussion of various influential factors relating to PMRs was used as a premise for developing an ideal reactor. The aim of this work was to resolve potential divergences that occur during PMRs design as most real reactors do not conform to the idealized fluid dynamics. Lastly, the application of PMRs is evaluated, not only in relation to the removal of endocrine-disrupting compounds (EDCs) from wastewater, but also in dye, oil, heavy metals, and pesticide removal.

## 1. Introduction

Photocatalytic membrane reactors (PMRs) are a promising technology in both academic research and the water industry. PMRs employ a synergistic approach in which the membranes combine photocatalysis and molecular sieving to overcome the technical and functional limitations of the one while leveraging the capabilities of the other [1,2,3]. Many benefits in terms of output and performance can be realized when heterogeneous photocatalysis is combined with membrane processes [4,5]. For instance, PMRs have the potential to become a green technology due to their low operating temperatures, use of chemically stable photocatalysts, and capacities to function in a continuous mode and to operate as single units, whereby photodegradation reactions, photocatalyst recovery, and product separation occurs in those units [6]. The pollutant removal mechanism in PMRs is initiated by the bulk diffusion and adsorption of pollutants onto the photocatalyst surface. After absorption, the photocatalyst undergoes photo-excitation induced by radiation. As a result, electrons are transferred from the valence band to the conduction band, leaving holes in the valence band. The generated electrons and holes then migrate to the surfaces of the photocatalysts to engage in oxidation–reduction reactions. In these reactions, the electrons and holes react with hydrogen ions and water molecules to produce radicals. These radicals are responsible for the partial or complete degradation of the adsorbed pollutants. The resulting products are then separated by the membrane through molecular sieving. Comprehension of the mechanical aspects of the PMR process necessitates an understanding of the fundamentals of photocatalysis and identification of the parameters that influence PMRs [7]. PMRs can be configured in numerous ways that influence photocatalytic system performance and offer possible solutions to issues such as catalyst activity, fouling management, selectivity, and membrane rejection [8,9,10]. This review seeks to provide recommendations and standards for the development of PMRs based on chemical engineering reactor design concepts. The review is built on the perception of readers that are more congenial with PMR reaction dynamics, fluid hydrodynamics, and architectural design. We begin by discussing the fundamentals of two main reactor types: slurry reactors and immobilized PMRs. The distinguishing properties of these PMRs are examined and emphasized to give clarity on their principal operations and different configurations, including the effects of each configuration on PMR operations. The dependence of PMR performance on membrane configuration, membrane rejection capabilities, photocatalyst loading capacity, light source, feed water quality, residence time, and reactor geometry is discussed. The goal is to use these criteria as a baseline for use in the development of efficient PMRs. As most PMRs are employed at a laboratory scale, the review goes on to address the potential challenges and gaps hindering the upscaling of PMRs. Lastly, the potential applications of PMRs in dye, oil, heavy metal, and pesticide remediation are explored.

### 1.1. Factors That Affect PMRs

While the potential of PMRs has been well documented, there are still several challenges relating to membranes used and photocatalytic reactions as well as the configurations of reactors that impede their practical application on an industrial scale [11,12,13]. Regarding membranes, membrane stability and fouling are two of the greatest challenges. The integrity of the membrane can be negatively affected by UV light, which is known to have a detrimental effect on chemically unstable polymeric membranes [13,14,15]. The use of solar or visible light instead of UV light can minimize the damage to polymeric membranes; alternatively, inorganic membranes may be used instead, since inorganic membranes can withstand UV light and resist oxidation by hydroxyl radicals [3]. In the case of slurry-based PMRs, the effectiveness of membrane separation can also be affected by the clogging of photocatalysts on membrane surfaces or inside membrane structures. Polymeric membranes, unlike ceramic membranes, are more susceptible to wear because of the abrasiveness of photocatalysts [16,17,18]. As a result, a new class of membranes with features such as chemical permanence and stability, as well as the ability to withstand harsh chemical and physical damage, is critically needed to address the challenges outlined above.

While fouling is not as severe in PMRs as it would be in a single-membrane system, foulant deposition on membrane surfaces remains a concern during the extended operation of PMRs as well as other membranes [19,20,21,22,23]. The main fouling process in PMRs is the formation of a photocatalyst cake layer on the membrane surface and tiny amounts of adsorption fouling produced by organic pollutants [24,25,26]. Additionally, some inorganic ions in the influent and degradation byproducts may bind to the photocatalyst immobilized on the membrane, leading to the development of a fouling layer [27,28]. Fouling reduces membrane permeate flux and increases pressure-related operational costs, as microfluidic resistance increases with the growth of the fouling layer. Regarding photocatalytic reactions and reactor design, reactor design works in concert with reaction kinetics. There are currently no verified kinetic models that enable the design of appropriate PMRs without the employment of costly and time-consuming traditional empirical models [29,30,31,32]. Hence, the development of PMRs necessitates a thorough understanding of kinetic models that account for all the process variables.

Up to now, there has been far too little interchange of knowledge and ideas about photocatalytic reaction kinetics. The application of reaction kinetics in the design and optimization of PMRs is crucial. This includes not only fine-tuning the mixing rates or fluid dynamics to meet the mass transfer requirements, but also the determination of the system’s optical properties, which are required to calculate the reactor volume for a given light intensity [33,34,35]. The configuration of the reactor (suspended or immobilized) affects these dynamics. In suspended PMRs, fluid dynamics can change photocatalyst distribution and aggregation state, deterring light usage and reaction kinetics [36,37,38]. The features of the fluid flow (laminar or turbulent) in immobilized PMRs, on the other hand, can decrease the mass transfer of pollutants to the photocatalyst surface. Knowledge of reaction kinetics and fluid dynamics must be discussed to accomplish an economically feasible conversion in PMRs, as modifying operating conditions alone may not be enough to make the process economically feasible and stimulate industrial application. As things stand, the PMRs with the most promising results are often designed at a laboratory scale and rarely replicated at an industrial scale, as operating conditions, such as flow rate, energy, pressure, and light intensity requirements, would alter with reactor capacity [39,40,41]. Therefore, conducting pilot studies is necessary and is recommended for the evaluation of feasibility, scaling factors, unpredictable results, and many other factors relevant to the implementation of a viable full-scale commercial process.

Another pertinent issue that affects the entire water treatment research area, including PMRs, is the use of simulated feed water for investigations rather than actual wastewater samples [42]. Wastewater is characterized by the presence of inorganic salts, suspended solids, and natural and dissolved organic matter. The ultimate performance of the PMRs would be considered inaccurate, when correlated with the real wastewater remediation. Evaluation of designed PMRs should be performed using real wastewater samples collected from water/wastewater treatment works as wastewater particulates have an impact on PMR performance [43]. This issue can be addressed by examining how each wastewater (WW) particulate affects the performance of PMRs. The suggested approach will make the testing of PMRs more realistic, allowing for easier upscaling of the process.

Regarding the exploitation of UV light, a remarkable amount of energy is consumed which accounts for most of the total cost of operations [3,15,44]. High operational costs have made the employment of PMRs economically unfeasible and have been a major stumbling block in PMR industrialization. However, energy costs can be lowered by developing solar-driven catalytically active materials with high activities and stabilities, making PMRs a cost-effective and sustainable technology. Finally, as things stand, the PMRs with the most promising results are often designed at the laboratory scale and are rarely replicated at an industrial scale, as operating conditions, such as flow rate, energy, pressure, and light intensity requirements, alter with reactor capacity [44,45,46]. Therefore, conducting pilot studies is necessary and is recommended for the evaluation of the feasibility, scaling factors, unpredictable results, and many other elements relevant to the implementation of a viable full-scale commercial process.

### 1.2. The Configuration of Photocatalytic Membrane Reactors (PMRs)

The configuration of PMRs governs the principal operation of PMRs and predicts the number of unit operations to be employed in the PMR water treatment process [5,42]. Table 1 summarizes some of the different PMR configurations that have been applied in water and wastewater treatment. PMRs differ from traditional photocatalytic reactors in that membranes are used to keep the photocatalysts confined in the reaction environment [43]. The two major configurations of PMRs are PMRs in which the photocatalysts are suspended and PMRs in which the photocatalysts are immobilized [44]. These configurations are discussed in the following sections.

#### 1.2.1. PMRs with Suspended Photocatalysts

PMRs with suspended photocatalysts are those in which the photocatalysts float freely within the confines of a vessel. This reactor type can be further classified into split-type and integrative PMRs [45,46,47,48]. Integrative PMRs are reactors wherein a membrane is immersed in a chamber containing suspended photocatalysts. Only two streams exist in this configuration: the feeding stream and the permeate stream [49,50]. Typically, the solution is abstracted from the outer to the inner side of the membrane under slight negative pressure, while the photocatalyst is intercepted on the outer membrane surface, ensuring that the photocatalyst content is constant in the photocatalytic tank [51]. Generally, because the hybrid system minimizes pipe length, head losses, and occupancy area, the integrative arrangement is more advantageous in terms of reducing operational costs [52]. Owing to the increased contact between catalysts and pollutants, generally suspended PMRs are widely employed in water treatment to improve mass transfer and photodegradation performance. The drawbacks of this type of configuration include high operating costs, the susceptibility to photocatalyst-induced membrane fouling, as well as damage by UV irradiation. Nguyen et al. [53] studied the removal of diclofenac utilizing an integrative cylindrical photocatalytic membrane reactor (PMR) with suspended N-TiO_2_ (Table 1). The reactor vessel had an immersed tubular microfiltration (MF) ceramic membrane encircled by five visible lamps of 50 W (420–720 nm), as illustrated in Figure 1. A suction pump was used to drive treated water towards the collection sample point. The permeate flux was measured after every 4 h of operation, and it was found that the permeate flux decreased within the initial 8 h of operation, achieving up to 50% diclofenac removal. The drastic decline in permeate flux was ascribed to the clogging of N-TiO_2_ on the membrane surface.

Similarly, Fernandez et al. [54] submerged an ultrafiltration (UF) hollow fiber membrane module with a filtering area of 0.047 m^2^ in a suspended TiO_2_ PMR for the removal of trihalomethanes. The membrane module’s permeate was coupled to a reverse peristaltic pump which performed semi-continuous filtration and backwashing operations in 4.5 min and 0.5 min cycles, respectively. As a result, 86% pollutant removal was achieved when operating at a TiO_2_ concentration of 0.5 g.L^−1^ (Table 1). To improve the performance of integrative PMRs, Plakas et al. [55] examined the effects of radiant power vs. hydraulic retention time. A fully automated PMR was used for the degradation of diclofenac (Table 1). The PMR components consisted of a hollow fiber UF membrane, a 52 W UV-C power lamp, and suspended TiO_2_ nanoparticles. The disseminated photocatalyst was completely retained by UF membranes, and pollutant removal ranged from 56% to 100%. However, the removal of total organic compounds (TOCs) was just 52%. Moreover, UV-C radiant power per unit volume boosted diclofenac elimination by 20%, whereas the hydraulic retention time (HRT) had no effect.

Szymanski et al. [56] installed a submerged polypropylene membrane in a PMR to study the influence of feed water on the removal of ketoprofen. The membrane was characterized by inner and outer diameters of 1.8 and 2.6 mm, respectively, and a membrane effective outer surface area of 0.0196 m^2^. TiO_2_ nanoparticles were used for the degradation of ketoprofen driven by light from UV lamps (Table 1). Ketoprofen removal solely depended on the initial concentration in the feed water, and complete degradation was achieved when the initial concentration was 10 mg.L^−1^ after running for 5 h.

**Table 1 membranes-12-00745-t001:** Summary of the different PMR configurations used in the degradation of organic pollutants.

PMR Configuration		Radiation Source	Photocatalyst, Membrane	Target Pollutant	Performance	Highlights	Ref
Suspended	Integrative	Visible	N-TiO_2,_ MF	Diclofenac	84.18%	- Addition of H_2_O_2_ enhanced the degradation of diclofenac - A dense cake layer of photocatalysts and pollutants was formed on the MF membrane	[53]
		UV	TiO_2,_ UF	Trihalomethanes	86%	- Hydrophobic organic particulates in the model water were absorbed into the membrane causing membrane fouling	[54]
		UV	TiO_2,_ UF	Diclofenac	100%	- pH of the feed water had a significant effect on the performance; high degradation was achieved in acidic conditions	[55]
		UV	TiO_2,_ UF	Ketoprofen	86%	- Thermal conductivity of the TiO_2_ improved the membrane permeate flux	[56]
	Split system	UV	ZnO, NF	Congo red	100%	- Significant numbers of ZnO photocatalysts were retained by the membrane and they were reused in the continuous process	[11]
		UV	TiO_2_, MF	Tannic acid	96%	- Improved membrane anti-fouling properties	[4]
		UV	TiO_2,_ MF	Azo dye Acid Red 1	<90%	- Performance was influenced by initial dye concentration. - Pseudo-first-order kinetics could not describe the reaction system	[12]
		UV	TiO_2,_ UF	Diclofenac	56%	- Hydraulic residence time had an insignificant effect on the performance	[40]
		UV	TiO_2,_ UF	Ibuprofen	100%	- No significant influence of operation mode was observed - Flux was recovered by cleaning with HCI	[37]
Immobilized		UV	TiO_2_, ZrO_2_ active layer on Al_2_O_3_ support, UF	Para-chlorobenzoic acid	0.088 min^−1^ removal rate	- Low kinetic rates were due to ions present in the feed water	[57]
		UV	Ag-TiO_2_ coated on Al_2_O_3_ porous membrane	Rhodamine, *E. coli*	1.007 mg m^−2^h^−1^, 7-log *E. coli* removed	- Antibacterial and photocatalytic properties of TiO_2_ were enhanced by Ag	[38]
		UV	LiCl-TiO_2_-doped PVDF, UF	Humic acid	90% humic acid rejection	- Improved rejection and membrane fouling properties	[44]

KEY: PVDF—polyvinylidene fluoride, BSA—bovine serum albumin, MF—microfiltration, UF—ultrafiltration, NF—nanofiltration, UV—ultraviolet.

As membrane fouling is a major issue in relation to the use of integrative PMRs, Gupta et al. [58] studied aeration and membrane oscillation as potential fouling control measures. The PMR was developed using TiO_2_ nanoparticles and 0.22 µm polyvinylidene fluoride (PVDF) flat sheet membrane. Aeration and oscillations were developed with a sparger and variable speed motor, respectively. Membrane oscillation performed up to 10 times better than aeration because of high shear rates. Despite superior oscillation performance, the membrane was susceptible to fouling by particles of about 0.1 µm.

To overcome UV irradiation-induced membrane damage and damage caused by reactive species, split-type PMRs are often used. Split-type PMRs (Figure 2a,b) are reactors in which photocatalysis, and membrane separations occur in two separate units. Normally, after photocatalysis, the reactive mixture is transferred to the membrane module for the separation and recovery of photocatalysts.

Hairom et al. [11] developed a split PMR with a suspended ZnO nanoparticle reactor unit and a flat sheet stainless steel membrane unit for the degradation of Congo red dye from wastewater (Table 1). A peristaltic pump was used as the driving force for the water matrix and 6 bar transmembrane pressure was achieved. The photodegradation efficiency was 70% and after nanofiltration (NF) the permeate was 100% free of Congo red dye.

#### 1.2.2. PMRs with Immobilized Photocatalysts

The other configuration for PMRs is that in which photocatalysts are immobilized on an inert support such as a glass substrate, to which a membrane module is attached for the separation of photocatalytic oxidation products [60,61,62,63]. The photocatalyst may in some instances be embedded in a support membrane that allows the photocatalyst to be separated from the effluent, preventing secondary pollution and photocatalyst losses [64]. The support membrane can act as a selective barrier for the removal of contaminants and as a support for the photocatalyst. The most commonly used membranes for this application are polymeric membranes and ceramic membranes [65,66,67,68,69,70,71]. Of the two types of membranes, polymeric membranes are frequently used due to their configurational flexibility, pore size range, and scalability. However, polymeric membranes are plagued by longstanding operational concerns, such as swelling, which reduces their lifetime and lowers their selectivity [72]. In addition, polymeric membranes are often sensitive to oxidation by hydroxyl radicals. As a result, due to their chemical resistance, ceramic membranes are preferable to their polymeric counterparts in catalyzed processes. Ceramic membranes allow for the combination of separation and chemical reactions, resulting in increased process efficiency with great mechanical stability and extremely high hydraulic permeability [71,72].

Immobilized PMRs are typically operated in one of two modes: dead-end flow or crossflow [32]. In dead-end mode (Figure 3), the feeding stream is pumped through the membrane to produce a cleaner permeate [71], whereas in crossflow mode, the contaminated stream is pumped to the coated side of the photocatalytic membrane (i.e., the active side) and flows in parallel with the surface of the membrane. The permeate moves through the membrane and the retentate, which contains higher concentrations of pollutants, is recycled back into the feed tank for further processing [72]. It must be noted that insufficient contact between molecules, photocatalysts, and light, along with the absence of a recycling stream, will often mean low photocatalytic efficiency. A laboratory-scale PMR (Figure 4) operating in crossflow mode was developed by Song et al. [44]. The developed system had a recycling stream and, as a result, high pollutant removal (Table 1) and low membrane fouling were achieved with this configuration. Some challenges presented by immobilized PMRs include loss of photoactivity, difficulty in irradiating the membrane surfaces, deactivation, and the washout of catalytic nanoparticles [72,73].

Janssens et al. [57] constructed a split PMR with a membrane module configured in crossflow mode. The reactor components consisted of a mercury lamp, a double diaphragm pump, a suspended TiO_2_ nanoparticle reactor unit, and a tubular ceramic membrane unit (with a ZnO active separation layer on an Al_2_O_3_ support). Water matrix circulation and pressurization were achieved at 3 bar. The innovative PMR was used to remove the drugs cyclophosphamide, 5-fluorouracil, and capecitabine from secondary effluent (Table 1). The photodegradation process followed pseudo-first-order kinetics with rate constants less than 0.03 min^−1^. Low photocatalytic activities were attributed to competition for absorption of UV light between water particulates and photocatalysts, and the oxidative radicals were probably scavenged by natural organic matter present in the secondary effluent wastewater.

Salehian et al. [73] employed a crossflow PMR composed of visible light LED lamps and a polysulfone/H_2_O_2_-g-C_3_N_4_ membrane module. An air compressor was used to increase oxygen concentration in the reaction environment. The reactor was operated at an applied pressure of 2 bar, achieving 412.1 Lm^−2^h^−1^bar^−1^ hydraulic permeability. Dissolved oxygen improved the self-cleaning properties of the membranes through photocatalysis, and the crossflow mode minimized the build-up of the foulant layer on the membranes. Hence, crossflow filtration is believed to be more suited to large-scale and continuous operations since dead-end flow suffers from membrane fouling because of the accumulation of separated substances on membrane surfaces [74,75]. In crossflow mode, the tangential feeding flow eliminates deposited substances on the membrane surface, minimizing the clogging of molecules in or on the membrane structure. However, overall performance is influenced by a variety of factors, such as the type of photocatalyst used, feed-water properties, light source, residence time, and membrane rejection.

## 2. Modeling of PMRs’ Operational Parameters

The modeling of parameters enables designers to develop PMRs which operate at optimum levels, giving optimum performance. Operation under extreme conditions may reduce the efficacy of the process. Such operational parameters include trans-membrane pressure, pH, aeration, inorganic particles in water matrices, and crossflow velocity [76,77,78]. In this review, the focus is on the parameters that influence the membrane processes, photocatalysis, and reactor geometry, and these operational factors are discussed in detail in the subsequent sections.

### 2.1. Flat vs. Cylindrical Membranes

Membrane configuration refers to the geometry and position in space of the membrane in relation to the fluid flow. Commercially, flat sheet and hollow membrane configurations are the sorts most often used in PMRs. Depending on lumen diameter, hollow membranes can be classified as fiber, tubular, or monolithic membranes [78]. Figure 5 shows the diagrammatic representation of flat sheet, tubular, and monolithic membranes. It is possible to configure tubular and monolithic membranes into multichannel configurations, unlike flat sheet membranes. However, ceramic monoliths have a high membrane surface-to-volume ratio compared to tubular elements of the same length [79,80]. Recently, the development of monolithic and tubular ceramic membranes has become a subject of interest owing to their high hydraulic permeability, narrow pore size distribution as compared to polymeric membranes, erosion resistance, high flux per unit membrane surface area, and durability [46]. Azrague et al. [46] examined the performance of a flat sheet membrane vs. a hollow fiber membrane in the treatment of turbid water in PMRs. Compared to its flat sheet membrane counterpart, the hollow fiber membrane was more prone to fouling. Hollow fiber membranes have higher surface areas than flat sheet membranes; hence, the reactor volume is substantially smaller, which is advantageous for industrial applications.

There are many benefits related to investigating the performance and economic feasibility of flat sheet vs. hollow fiber membranes. Data were continuously collected over 2 years of operations. A performance evaluation was based on suspended solids and color removal. Within 6 weeks of operation, the flat sheet membrane had already clogged, and the hollow fiber membrane process ran for up to 16 weeks without clogging [79]. The hollow fiber membrane had lower maintenance costs than the flat sheet membrane. The flat sheet membrane had high chemical consumption during wash time, again showing high susceptibility to fouling [79].

Moreover, hollow fiber membranes may be operated in an outside–in or inside–out configuration. With the advancements in membrane techniques, outside–in hollow fiber membrane modules are said to be more resistant to fouling than inside–out ones. Although both modules can meet the permeate quality requirements, Xu et al. [81] discovered that an outside–in module with an air-enhanced backwash consistently outperformed an inside–out module with a chemical-enhanced backwash. As a result, outside–in hollow fiber membranes are used more often in PMR systems.

### 2.2. Membrane Solute Rejection Properties

Rejection is a useful feature that plays a role in the selection of membranes for wastewater treatment. Ideally, the membrane pore size should be smaller than the size of the target pollutants to be rejected in order to achieve high rejection [82]. Furthermore, regulating parameters such as pH, concentration polarization processes, and residence time may result in the retention of contaminants [83,84]. The Donnan effect demonstrates the importance of surface charges on membrane rejections. According to the Donnan effect, for instance, if the charges between the membrane and the targeted pollutants are of the same sign, the molecules would be repelled by the membrane; however, if the membrane and pollutants bear dissimilar surface charges, attractive interactions prevail [84,85]. Repulsive interactions increase rejection, while attractive interactions lower rejection due to the high attraction of the targeted compound, leading to pollutant adsorption and thus lower removal.

Mahlangu et al. [83] studied NF membrane rejection properties against NaCl and carbamazepine pollutants. Membrane rejection for NaCl was 47%, and the low rejection value was due to low membrane zeta potentials. For carbamazepine, 60% rejection was achieved, and rejection was due to steric hindrance. Several other researchers have demonstrated the influence of membrane zeta potential, pollutant charges, pore size, and steric hindrance on membrane rejection [86,87,88,89].

### 2.3. Photocatalysts Loading Capacity

Variation in photocatalysts’ loading capacities can either have negative or positive effects on the performance of PMRs. In the case of suspended PMRs, an increase in the concentration of photocatalysts decreases photocatalytic activity. A high photocatalyst concentration makes the reactive solution opaquer and therefore reduces the ability of photons to penetrate the solution. Additionally, the use of elevated levels of photocatalysts can also result in the agglomeration of photocatalysts, which reduces the total surface area of the photocatalysts [90,91,92,93,94]. As a result, several researchers have aimed to optimize photocatalytic loading capacity. These efforts include the work of Adan et al. [95], who investigated the synergistic effect of photocatalyst loading on methanol degradation. The TiO_2_ loading and pore size of the membrane had a noticeable effect on the efficiency of the process and on transmembrane pressure. The optimal photocatalyst loading capacity obtained was 8.5 g L^−1^, with a transmembrane pressure of 1.5 bar to 2 bar. The best TiO_2_ loading was selected due to a trade-off between light absorption and pressure. Elsewhere, to remove acid red dye from water, Wang et al. [96] used an immobilized TiO_2_ porous stainless steel membrane. The optimum TiO_2_ loading capacity was found to be 0.29 g L^−1^, with a range of 0.03 to 0.44 g L^−1^. The degradation rate increased when the loading of the catalyst was increased but stabilized and remained constant when optimum photocatalytic loading was attained. Szymanski et al. [56] used a TiO_2_ ultrafiltration membrane-based PMR and reached a similar conclusion—that photocatalyst loading capacity does indeed directly affect the performance of PMRs. The study revealed that membrane fouling could be improved by increasing TiO_2_ loading from 0.5 to 1.5 g L^−1^.

### 2.4. Characteristics of the Light Source

Wavelength and light intensity are important parameters with a noticeable effect on photocatalysis. Light can be defined as a narrow frequency band of electromagnetic waves [97]. There are two main sources of light relevant to PMRs: solar and artificial light (lamps and light-emitting diodes (LEDs)). Light propagation in PMRs is influenced by two processes: absorption (which depends on the intensity of the electromagnetic field) and scattering (which depends on the refractive behavior of the particulates in the reactive mixture) [98,99]. Maxwell’s computational equation is one of the models that can provide insights into light-scattering properties in a system. Further, the rational selection of the light source has a significant impact on PMR reaction kinetics and on the energy utilization ratio.

Regarding to the wavelength, most photocatalysts can be activated exclusively by UV light [3]. This limits their application as UV lighting system has a high energy input requirement and in case of natural light, UV light constitutes only approximately 5% of the solar spectrum [100]. The excessive energy input associated with the use of UV light results in the formation of byproducts instead of the complete mineralization of pollutants and it has high operational energy costs [92,101]. The use of visible light as a light source can help to mitigate the mentioned issues. Mendez et al. [102] doped Au with TiO_2_ to make visible light-sensitive nanoparticles via a multi-photon absorption mechanism. The absorption edge energy of TiO_2_ changed from 3.15 eV to 2.98 eV after Au doping, indicating that the absorption peak was red-shifted, and the Au doping improved visible light absorption. Muleja et al. [103] doped calcinated TiO_2_ nanoparticles with cobalt to produce visible light-sensitive photocatalysts. The synthesized calcinated Co/TiO_2_ nanoparticles had a bandgap of 1.86 eV and an absorbance range of 400 to 800 nm. Tshangana et al. [104] also reduced the band energy value of photocatalysts by combining ZnO with graphene quantum dots. The bandgap value was reduced from 2.98 to 2.61 eV. Moreover, other authors have also proven that metal doping and combining photocatalysts with other metal oxides can produce visible light-sensitive catalysts [63,105,106].

When light intensity increases, photocatalytic reaction rates increase as well, if intensities do not exceed the optimum thresholds, which vary according to photocatalyst type and other factors [107]. Exceeding the optimum threshold value does not have any further effect on reaction kinetics. Planck’s equation (Equation (1)) demonstrates the proportionality between the energy of photons and light frequency [108].
(1)$$E_{Photon}=hv$$
where $E_{Photon}$ is the energy of photons, h is Planck’s constant, and v is the frequency of light.
(2)$$E_{photon}=KE_{electrons}+\emptyset$$
where $KE_{electrons}$ is photoelectron kinetic energy and ∅ is the minimum energy required to induce emission of electrons.

Hence, if the energy of photons increases, photocatalysts will have enough energy to induce the photoemission of electrons, as described by Equation (2). Wang et al. [51] used integrative PMR with suspended C-N-S tri-doped TiO_2_ to decompose carbamazepine (CBZ) under vis-LED irradiation, and the degrading effects of CBZ increased from 28% to 68% with an increase in vis-LED units from 60 to 240 nm. These results showed that reaction rates increase with light intensity up to an optimum value.

### 2.5. Residence Time

Residence time also determines the efficiency of the photocatalytic degradation process. A longer residence time provides enough time for contact between pollutants and photocatalysts. However, since PMRs should be able to offer high permeate flux, a good compromise should exist between residence time and permeate flux. Figure 6 shows how naproxen concentration removal depended on residence time. Wang et al. [109] achieved 87.7% naproxen removal by extending the UF membrane residence time to the same removal rate as that of the NF membranes.

### 2.6. Initial Pollutant Concentration

Feed concentration stipulates the number of target pollutants per a given volume. Initial pollutant concentration depends on the source of wastewater; for instance, sulfamethoxazole has been measured in effluents of wastewater treatment plants (WWTPs) in a concentration range of 0.6 µg L^−1^ up to 34 µg L^−1^ [110]. Carbamazepine (CBZ) residues in wastewater have been reported in KwaZulu-Natal province [111] in a concentration range of 30 to 340 ng L^−1^ [112]. In a review by Madikizela et al. [113], nonsteroidal drugs, such as ketoprofen and diclofenac, were detected in water and sediment of the Umgeni River (SA) at concentrations of up to 14.4 µg L^−1^ and 26.5 µg L^−1^, respectively.

Moreover, the Langmuir–Hinshelwood kinetic model (Equation (3)) is an expression that describes the relationship between photodegradation rate and pollutant concentration [51].
(3)$$r=\frac{\mathrm{dC}}{\mathrm{dt}}=\frac{K_{r}K_{\mathrm{ad}}C}{1+K_{\mathrm{ad}}C}$$
where k_r_ is the intrinsic rate constant, C is pollutant concentration, and K_ad_ is the adsorption equilibrium constant.

Further, Equation (3) can be simplified to Equation (4) when the concentration of the contaminant of interest is low. Under such circumstances, the rate of reaction will depend on time rather than pollutant concentration.
(4)$$\ln\frac{C}{C_{o}}=-k_{r}k_{\mathrm{ad}}t=-k_{\mathrm{app}}t$$
where C and C_O_ are initial and final pollutant concentrations, respectively; t is time; and K_app_ is the apparent rate constant of a pseudo-first-order reaction.

Within a range, a high initial pollutant concentration speeds up the photocatalytic degradation reaction. High concentrations increase the chances of successful collisions between target pollutants and oxidative species [50,74,114,115,116]. On the other hand, a high initial concentration can also slow the photocatalytic degradation rate. This is because excessive pollutant concentrations can alter solution opacity, causing pollutants to absorb light rather than the photocatalyst, and this lowers photocatalytic effectiveness. At elevated concentrations, the contaminants are more likely to occupy the active sites of the photocatalysts, thus lowering photocatalytic efficiency. Hence, the use of pre-treatment processes, such as sedimentation, clarifiers, adsorption, and many others, can help improve PMR degradation kinetics. Sakarkar et al. [115] studied the influence of the concentration of remazol turquoise blue dye on photodegradation performance using TiO_2_ immobilized on a polyvinylidene fluoride (PVDF) membrane. The dye concentration varied from 50 mg L^−1^ to 200 mg.L^−1^. The TiO_2_/PVDF membrane dye removal capacity decreased with an increase in initial dye concentration. The reduction in dye removal efficiency showed that high dye concentrations increased light attenuation and hence reduced the number of photons available for photocatalysis on the membrane surface. On the other hand, the high dye concentration also increased membrane fouling resistance. Elsewhere, Iqbal et al. [116] used a sulfamethoxazole concentration range of 2.5 mg L^−1^ to 20 mg L^−1^ to investigate the influence of concentration on sulfamethoxazole photocatalytic degradation performance. A maximum of 97% and a minimum of 59% degradation were achieved using 2.5 mg L^−1^ and 20 mg L^−1^, respectively. This showed that an increase in the initial concentration of sulfamethoxazole decreased the degradation performance. Tshangana et al. [117] also investigated the relationship between initial concentration and photocatalytic process efficiency. Graphene quantum dots were used to degrade an azo dye in a concentration range of 15 mg L^−1^ to 50 mg L^−1^. Degradation performance decreased from 98% to 60% with the increase in initial dye concentration.

### 2.7. Reactor Geometry

Reactor geometry refers to the shape or dimensions of reactor components and their arrangement in space. Reactor thickness is one of the geometrical features that affects the radiation of externally radiated slurry PMRs. Thin- and thick-slurry PMRs are illuminated differently. Thin reactors are homogeneously illuminated compared to thicker reactors. Several authors have demonstrated that thick reactors are subjected to the formation of dark zones [60,69]. As a result, the optical thickness of a photoreactor is commonly acknowledged as the most important parameter, considering both the geometrical thickness and photocatalyst concentration of the photoreactor. Channel diameter is another geometrical parameter of importance. A design should offer a good comprise between optimal channel diameter and operating pressure to minimize pressure-dropping along the membrane. A pressure drop results in a static volume inside the flow channels, and this increases process energy requirements as more energy is required for pumping to overcome the frictional forces caused by fluid viscosity or by the wall of the reactor vessel. The frictional forces can be determined using the Darcy friction factor (*f*) mathematical model (Equation (5)) for tubular reactors [118]. The model relates pressure loss and dynamic pressure (resembling the kinetic energy in the fluid flow), considering the diameter and length of the reactor.
(5)$$f=\frac{∆P}{\frac{1}{2}\rho v^{2}}\times \frac{d}{L}$$
where f is the Darcy friction factor, ∆P is dynamic pressure, d is vessel diameter, L is vessel length, ρ is the fluid density, and v is fluid velocity.

Tugaoen et al. [119] developed a polyvinyl chloride cylindrical reactor with a channel diameter of 1.9 cm and a length of 18 cm (Figure 7). The feed solution was driven by a peristaltic pump at a rate of 5 mL min^−1^, leading to a hydraulic retention time of 10 min within the reactor. Athanasiou et al. [13] used multichannel monolithic UF membranes with 2.9 cm diameters and 123 cm lengths. The design was intended to minimize the static reactor volume and eliminate energy losses due to frictional force by maintaining the reaction in fluid flow at a minimum.

Several kinds of research have been performed regarding the optimization of PMRs, but the issue of upscaling PMRs remains. The following subsection describes the challenges and gaps in current PMR design hindering the upscaling of PMRs.

## 3. Gaps and Challenges in Current PMR Design

Despite the benefits of adopting PMRs, there are still certain challenges to be resolved. The upscaling of photocatalytic reactors is one of the most difficult topics in photocatalytic reaction engineering. In simulating large-scale PMRs, many approaches have been used in the literature. The presence of user-friendly mathematical models that preserve the fundamental aspects of rigorous models while being easier to utilize for scale-up and design objectives can help with upscaling [120]. A design’s resilience stems from the development of a mathematical model that connects material and mass balances to the following sub-models: a fluid dynamic model, a radiation emission model, a radiation absorption model, and a kinetic model [121]. The performance of PMRs is highly dependent on the light sources and light dispersion inside the reactor volume. Consequently, consideration must be given to the optimization of the irradiation process. The precise determination of a radiation field is difficult due to a lack of proper radiation models, kinetic models, and design techniques [122,123], and the behavior of light within heterogeneous media and its influence on the local pollutant degradation rate are still poorly understood. Maxwell’s equations on the other hand can be utilized to explain radiation models. Maxwell’s equations provide a mathematical representation of the overall electromagnetic field in the presence of optical particles. The propagation of emitted radiation depends on the scattering properties of the particles present in the reactive mixture, and the absorption of the radiation depends on the radiation intensity. The radiative transfer equation (Equation (6)), which may be obtained from Maxwell’s equations, is then used to calculate the radiation distribution. Ripoll [124] discusses the derivation of the radiative transfer equation from Maxwell’s equations. By making absorption (A) or emission (B) functions the subject of Equation (6), radiation models (emission or absorption) can be calculated.
(6)$$\frac{{dI}_{\lambda ,\Omega }(s,t)}{ds}+\alpha_{\lambda }(s,t)I_{\lambda ,\Omega }(s,t)+\underset{\left(A\right)}{\underset{︸}{\delta \lambda (s,t)I_{\lambda \Omega }(s,t)}}\underset{\left(B\right)}{\underset{︸}{=j_{\lambda }^{e}(s,t)}}+\frac{\delta \lambda (s,t)}{4\pi }\int_{\Omega^{\prime }=4\pi }p(\Omega^{\prime }\to \Omega )I_{\lambda \Omega }(s,t)d\Omega^{\prime }$$
where s is a spatial parameter, t is time, $p\left(\Omega^{\prime }∆\Omega \right)$ is a scattering phase function, $\alpha_{\lambda }$ is a volumetric absorption coefficient, *e* is exponential functions, $\delta_{\lambda }$ is a scattering coefficient, and $I_{\lambda ,\Omega }\left(s,t\right)$ is spectral radiation intensity. Various modeling techniques have been proposed depending on the configuration of PMRs. For suspended PMRs, emphasis is placed on the correlation between radiation and mass transfer. The suspended catalyst is impacted by light scattering, which impairs the distribution of light. Modeling the radiation field in immobilized PMRs is easier, as the immobilization of catalysts minimizes light scattering. As a result, intrinsic and exterior mass transfer phenomena are given more attention [125].

In the design of PMRs, the Local Volumetric Rate of Photon Absorption (LVRPA) is an important aspect [126]. However, LVRPA has been approximated indirectly in the solution of the radiative transfer equation [127]. Accurately determining the LVRPA spatial distributions within PMRs guarantees improved design, despite the lack of accurate kinetic information derived from averaged photon absorption rates and the severe non-uniformities intrinsic to light propagation in scattering-absorption media.

Furthermore, even though light intensity can only be measured in certain subsets of physical space around solids, recreating radiation fields from such experimental data remains difficult. The Helmholtz equation [128] was employed to simulate light distribution inside solid media using two constant parameters: scattering and absorption coefficients. The Helmholtz equation has been examined in several applications, including impedance imaging and wave propagation, and scattering biological imaging [129]. However, besides the report by Vaiano et al. [130] in 2015, as far as we know, at the time of writing this review, the Helmholtz equation has not been used to describe the distribution of photons in the photocatalytic reaction zone.

The succeeding subsection elaborates on how reaction kinetics is intertwined with basic PMR design principles.

### 3.1. Reaction Kinetics

One of the key aspects of reactor engineering is reaction kinetics. Reaction kinetics allows qualitative or quantitative evaluation of the rate of reactions, as well as insight into the dependency of the rate of reactions on variables such as pollutant concentration, pH, temperature, and light intensity [130,131,132]. Understanding the kinetics of a reaction is essential for controlling the reaction and achieving the intended yield. Kinetic models such as the Langmuir model (Equation (4)) have been developed to solve rate constants that are required in reactor design. However, the assumptions used to evaluate reaction kinetics may not conform to real reactors, as perfect mixing and mass transfer cannot be achieved in non-ideal situations [133,134,135]. Consequently, undersized reactors with inaccurate residence times may be developed. Hence, for an economical and safe reactor design, precise kinetic rate expressions are required. Assidi et al. [134] conducted kinetic modeling using a reactor with a mass flow controller and a venturi. Low degradation performance was attained at high flow rates. In this design, mass transfer was the limiting step, and rate and adsorption constants were functions of radiation intensity but not of flow rate. Tisa et al. [33] developed a fluidized PMR with a reactor length of 10 m. The reaction conversion increased up to the equilibrium point with residence time, and the reaction rate increased across the reactor length, achieving a concentration decrease of 4.6 mol/dm^−3^.

On the other hand, rate data can be utilized to hypothesize a reaction’s kinetic sequence by proposing stoichiometrically consistent elementary steps. The rate expression is derived based on assumptions such as, a steady-state, a rate-determining step, and the most abundant reaction intermediate [135]. The suggested rate expression can only be deemed acceptable when its functional dependency resembles that of kinetic data. Further, even if the kinetic parameters closely match the data, there is no certainty that the solution accurately captures the real kinetic sequence. To have confidence in the kinetic model, numerous kinetic models must be developed and tested against one another.

However, for generating kinetic models for photocatalytic reactions, two approaches can be used: semiconductor and mechanistic approaches. These methods consider how electron–hole production and recombination affect reaction dynamics. A mechanistic approach establishes a kinetic model based on the law of mass action for both reactive species and electron–hole pairs. The recombination rate is determined using Equation (7).
(7)$$r_{rec}=k_{rec}\left[h^{+}\right]\left[e^{-}\right]$$
where $r_{rec}$ is the rate of recombination reaction, $k_{rec}$ is the recombination rate constant, $\left[h^{+}\right]$ is the concentration of positive charge carriers, and $\left[e^{-}\right]$ is the negative charge carrier concentration. There is an imbalance of produced charge carriers in the case of doped photocatalysts. This reduces recombination, and the rate of a photocatalytic reaction can be determined using Equation (8).
(8)$$r=\left[h^{+}\right]\propto I^{0.5}$$

For the semiconductor approach, electron and hole recombination is deemed insignificant. From the semiconductor approach, Nielsen et al. [136] developed a kinetic model (Equation (9)) based on the assumption that photovoltage, $V_{ph}$, is the driving force of the reaction rate.
(9)$$r\propto \frac{\alpha \emptyset_{0}e^{-\alpha x}}{k_{\mathrm{rec}}\left[h^{+}\right]{\left[e^{-}\right]}_{o}h\omega }$$
where α is the absorption coefficient, ∅ is photon flux density, and hω is photon energy. It is important to note that Equation (9) only applies to ideal inherent semiconductors with a uniform crystalline lattice.

### 3.2. Membrane Stability in PMRs

Membrane separation is a necessary aspect of PMR operations, and their long-term stability under continuous operation is still uncertain. The application of UV radiation may inevitably present operational challenges for polymeric membranes. In polymeric membranes, the chemistry of the polymer matrix is important in the determination of membrane stability. UV light tends to degrade polymeric membranes, unlike inorganic membranes [14,15,16]. Under UV radiation, smaller plastic particles (<200 nm) can be liberated from polymers, and tiny particles have a greater influence on biodiversity than bigger particles [124]. Sanches et al. [137] evaluated the resilience of polymeric membranes under UV light, both before and after coating with TiO_2_. Membrane stability was assessed using changes in hydrophilicity and particle analysis in water. In comparison to uncoated membranes, coated membranes were more stable. The photocatalytic characteristics of TiO_2_ nanoparticles gave the membrane photocatalytic capabilities which prevented morphological and chemical changes. However, the use of solar or visible light instead of UV light can minimize damage to polymeric membranes, as can, alternatively, opting for inorganic membranes, since inorganic membranes can withstand UV light and resist oxidation by hydroxyl radicals [3].

The structural or chemical integrity of membranes is not only affected by UV degradation; mechanical stress and extreme temperature may also contribute to membrane degradation. Carbon nanotubes doped with polyamide membranes showed an improved stability under higher temperature and UV irradiation [138]. The addition of carbon nanotubes improved the mechanical and chemical properties of the membranes by changing the lamellar orientation, viscosity, and dispersion of the membranes. Moreover, degradation of the polymeric membranes promotes the leaching of nanoparticles from the nanocomposite membranes, and this results in reduced antibacterial and antifouling properties. Zodrow et al. [139] reported a decline in membrane performance during operation, and it was observed that 10% of Ag nanoparticles had leached in the first filtration cycle. Wan et al. [140] reported that interactions between nanoparticle and additives can minimize the loss of nanoparticles through mitigation of nanoparticle aggregation. An Fe/Pd-doped poly(vinylidene fluoride) membrane was more stable after the addition of polyacrylic acid.

## 4. Design of an Ideal PMR

There are four major design considerations for an ideal PMR: throughput, performance, energy efficiency, and economical feasibility [141]. These factors must be considered to achieve the best design results. The design process usually starts with deriving design equations which depend on the mode of operation of the reactors: batch-wise or continuous flow (Equations (10)–(12)) [97]. Moreover, fluid hydrodynamics can also be used to differentiate these reactors; hence, reactors exist as batch reactors, continuous stirred tank reactors (CSTRs), and plug flow reactors (PFR)s [142,143].
(10)$$batch=\int_{nf}^{no}\frac{dn_{A}}{-rxV}\;$$
(11)$$CSTR=\frac{no-nf}{-r}\;$$
(12)$$PFR=\int_{nf}^{no}\frac{dn}{-r}\;$$
where no is the initial number of moles of the pollutant, nf is the final number of moles of the pollutant, r is the rate of the reaction, and V is the volume.

To solve the design equation, the rate of reaction for a catalytic reaction can be calculated using rate constants (k) obtained from the Arrhenius equation (Equation (13)).
(13)$$k=e^{\frac{-E_{a}}{RT}}$$
where $E_{a}$ represents activation energy, T represents thermodynamic temperature, and R represents gas molar constant. k can be substituted in Equation (14) to obtain the rate (r).
(14)$$\ln\left(\frac{d\left[A\right]}{dt}\right)=\ln k+n\ln\left[A\right]$$
where $\left[A\right]$ is the pollutant concentration and $\frac{d\left[A\right]}{dt}$ is the rate of the reaction. The Arrhenius equation is mostly employed as the standard evaluator of the intrinsic kinetics of photocatalysts. For instance, the determination of the catalytic performance of Cu_2_O nanoparticles in the oxidation of CO using the Arrhenius equation was reported by Bao et al. [143].

Batch reactors are vessels in which reactants are given a set amount of time to mix, react, and produce products [144]. For a CSTR model, the vessel permits the reaction mixture to flow in and out without being held for a set amount of time, resulting in steady-state operation and homogeneous mixing across the reactor volume. PFRs (also known as continuous tubular reactors) are cylindrical tubes through which pollutants pass and where reactions occur as the pollutants pass through [145]. The length of the cylindrical tube is a critical parameter to consider since it influences the reaction progress. The flow behavior of PFRs (Figure 8a) is characterized by a constant fluid velocity moving in a radial direction across the vessel’s cross-section [146,147]. Back-mixing does not occur within the plugs of fluid moving through the reactor. In CSTR and batch reactors, three-dimensional and recirculating turbulent flow patterns are generated by a revolving impeller [148,149]. The flow pattern might be radial or axial depending on the type of impeller used (Figure 8b,c) [149]. The liquid flow profile, on the other hand, must be consistent, otherwise the contact between the pollutants and the catalyst surface will be compromised, impairing photocatalytic degradation overall [150,151,152,153].

From a practical point of view, most real reactors do not conform to the hydrodynamics of ideal reactors, but designers strive to make real reactors as close to the ideals (batch, CSTR, and PFR) as possible [97,151]. The deviation from idealized models may be due to the presence of stagnant zones in reactors, fluid channeling, recycling streams, and bypass. The flow patterns in PMRs can be described using the Reynolds number (Re). For turbulent flow, $Sh\sim Re^{0.8}$, and for laminar flow, Sh~Re13, where Sh is Sherwood number (mass transfer coefficient). Alterations to fluid flow patterns may affect mass transfer phenomena, which can deter the efficient degradation of pollutants [152], and it must be noted that the Reynolds number also depends on reactor geometry. For instance, Rezaei et al. [153] operated an annular geometrical PMR in a turbulent flow regime at 2526 Re and with a flow rate of $7\times 10^{-5}m^{3}.s^{-1}$. The system achieved 74% phenol degradation efficiency. Chekir et al. [154] used a compound parabolic collector for the degradation of paracetamol. Paracetamol was efficiently degraded up to 99% when the reactor was operated at 1261 Re in a laminar regime. Careful consideration must be given when it comes to monitoring reactor flow patterns, as increases in Re values can only have positive effects on performance up to a critical point. Any increase above this point decreases performance [154,155,156].

Traditionally, batch reactors have been employed to explore conversion performance as well as reaction mechanisms at a smaller scale [156]. Batch reactors are distinguished by their simplicity of construction, long residence time, and ease of handling. However, batch operations suffer from low throughput [157]. With the development of PMRs as a feasible green alternative for water remediation, attempts to utilize continuous flow reactors have been made. Continuous flow reactors are preferred over batch reactors because of their higher process throughput and efficiency, simpler operations, and ease of scale-up [158,159,160]. To minimize the development of undersized PMRs, variables such as efficacy factor (the rationalized factor of actual to observed rate) and a rationalized factor of the radiated surface area of the catalyst to the specific reactor volume can be used. Hence, for a continuous flow reactor, the mass balance of the process can be expressed using Equation (15).
(15)$$QC_{0}-QX_{A}C_{0}=\left(-r_{A}ŋ\right)V_{R}ᴤ$$
where Q is volumetric flowrate, $C_{0}$ is inlet pollutant concentration, $X_{A}$ is conversion, $-r_{A}$ is reaction rate, ŋ is efficacy factor, $V_{R}$ is reactor volume, and ᴤ is the specific radiated surface area of the photocatalyst. The reactor volume of a well-mixed PMR can be derived using Equation (16).
(16)$$V_{R}=\frac{{\mathrm{QC}}_{0}-{\mathrm{QX}}_{A}C_{0}}{-r_{A}ŋᴤ}\;$$

For PFR, reactor volume can be calculated using Equation (17).
(17)$$V_{R}=\frac{{\mathrm{QC}}_{0}}{ŋᴤ}\int_{0}^{X_{A}}\frac{dX_{A}}{-r_{A}}\;$$

As an important reactor design aspect, energy consumption can be used as a baseline to measure PMRs’ performance. Energy efficiency can be defined using the electrical energy per order parameter (ω), which is given by Equation (18). Electrical energy per order can be used to find the optimum number of optical fibers to be used for maximum illumination [160,161].
(18)$$\;\omega =\frac{P_{light}}{Q\log\frac{C_{0}}{C}}\;$$
where C is the final concentration of pollutants and $P_{\mathrm{light}}$ is the electrical energy used by the light source. Tugaoen et al. [119] developed an optical fiber PMR system. The relationship between quantum efficiency, and the number of optical fibers was investigated by assessing different ratios of lamp-to-number of optical fibers (Figure 9). 1:1 scenario obtained the highest electrical energy per order parameter ω with a lower quantum efficiency.

Several designs have been proposed for the development of PMRs in the literature [46,162,163]. For example, Kanmani et al. [164] developed a solar-based PMR that uses photocatalytic solar energy collectors to produce an energy-efficient system. However, the design was limited by temporal variation in light intensity, which resulted in slow kinetics. Later, PMRs evolved into artificial lighting systems that employ standard lights. Chong et al. [165] developed a slurry annular PMR; although the PMR design was simple, it was difficult to recover the catalyst. Further, there was uneven light illumination through the reaction medium. To improve mixing and catalyst recovery, Yatmaz et al. [162] designed a spinning disc PMR with a single baffled disc rotating at 350 RPM. The system showed mass transfer limitation, which affected the overall performance of the reactor. Mirzaei et al. [163] optimized the spinning disc reactor by increasing the number of baffles from 4 to 10. The improved design by Mirzaei et al. [163] enhanced phenol degradation. Although the spinning disc design was promising, unfortunately, the design was hindered by high electricity costs and the requirements of a large area for the disc to be established within the reactor.

There are other important engineering design variables that need to be considered, such as the material used for construction, thickness, and reactor geometry [165,166,167]. Table 2 depicts the need to strike a balance between productivity and efficiency in designing an efficient PMR [49,50], where increased mass transfer has an impact on productivity. Mass transfer efficiency, on the other hand, is influenced by irradiation time, light intensity, catalyst load, and pH. An ideal reactor design should have a high photocatalytic space–time yield (PSTY) and a high space–time yield (STY) [49]. STY is used to estimate the quantity of pollutants that can be treated from 100 mmol L^−1^ to 0.1 mmol L^−1^. The reaction rate constant and the reactor’s hydrodynamic characteristics determine the STY value. While PSTY is found by dividing STY by the lamp power required to radiate 1 L, PSTY covers both mass transfer efficiency and illumination efficiency. For example, annular PMRs have low mass transfer rates and high illumination efficiencies, while micro-PMRs have high mass transfer and low illumination [50].

To address the abovementioned design limitations, the following reactor design criteria are proposed:

### 4.1. Material of Construction

The selection of a chemically and physically stable material is preferable when it comes to the construction of reactors [168]. Over the decades, reactor vessels have been constructed from glass stainless steel and/or plastic materials. Due to the nature of PMR operation, which requires light irradiation, the material for construction of a PMR should be penetrable by light to allow the activation of the reactions between pollutants and photocatalysts in the reactor. However, advancement in technology has made it possible to install the lighting system inside the reactor vessel. Tugaoen et al. [119] used polyvinylchloride (PVC) to construct a reactor casing. Fernandez et al. [54] employed Plexiglass to create an integrative PMR with suspended TiO_2_ to remove trihalomethanes from drinking water, achieving a pollutant removal rate of 86%. In a nutshell, the selection of the material for the construction of a PMR should be determined by reaction conditions and consideration of the following parameters: flow rate, temperature, pH, pressure, mass load, velocity, light (source, type, irradiation intensity), among others.

### 4.2. Thickness

The thickness of PMRs depends on features such as operation pressure [169]. An ideal thickness should be able to withstand the design pressure. This design concept can be formulated using the Hoop stress formula (Equation (19)).
(19)$$t=\frac{PR}{SE-0.6P}+t_{c}$$
where *t* is the vessel wall thickness, *P* is the design pressure, *R* is the inside radius of the vessel, *S* is the maximum allowable stress for the steel, *E* is the joint efficiency, and *t_c_* is the corrosion allowance.

The Hoop stress formula applies to all type of reactors (batch, CSTR, and PFR). What differentiates thickness values is the impact of parameters such as operating pressure, flow rate, the nature of the reactants, and so forth. As mentioned earlier, in Section 2.7, thickness also affects the radiation performance of reactors with an externally configured lighting system. Dutta et al. [156] developed a Taylor vortex reactor with an inner diameter of 0.0434 m and an outer diameter of 0.0523 m, giving a thickness of approximately 0.0111 m. However, the performance of this design was directly linked to the magnitude of the vortex and the amount of photocatalyst used. There is limited information on PMR thickness, but it is important to consider reactor thickness because a balance must be found between efficiency and durability.

### 4.3. Source of Light

Light has a direct impact on the performance and energy efficiency of PMRs. However, it is critical to ensure that the wavelength of light does not fall within the absorption range of pollutants. The photodegradation rate is also influenced by the distance between the reaction media and the light source. To achieve a consistent intensity and minimize dark zones, the light path length in the reaction fluid should be 20 to 50 nm [6,159,166]. Further, it is also important to understand the hydrodynamics of PMRs before considering various light source configurations. The heterogeneity of the process makes it difficult to design and optimize PMRs [169]. Computational fluid dynamic models can be used to study fluid dynamics by, firstly, defining the flow characteristics using the Eulerian–Lagrangian or the Eulerian–Eulerian approach. Among these approaches, the Eulerian–Eulerian is the most appropriate because observations (velocity, density, acceleration) are taken from a fixed region rather than following a fluid particle in motion and observing changes.

In terms of light supply, artificial light sources (lamps and LEDs) are preferred over solar radiation due to fluctuations in light intensity, low kinetics, and the complexity of the designs and components used in solar reactor systems [6,130,169,170,171,172,173]. In contrast to solar irradiation, artificial lights have a narrower emission spectrum which aligns with the UV and visible regions. LED lighting systems are lightweight, compact, rugged, and operate at low temperatures compared to artificial lamps. According to Haitz’s law, the performance of LEDs improves exponentially as photocatalysts, optics, and materials science technology advances [174]. On the contrary, the performance of artificial lamps is affected by thermal radiation that occurs in the lamp filament and energy loss incurred through the conversion of light energy [175]. Artificial lamps have a short-rated life span compared to LEDs and they are toxic, containing harmful substances such as mercury.

It is worth noticing that an efficient source of light produces uniform light intensity, minimizes thermal radiation during operation, possesses a narrow emission spectrum, does not emit a wavelength that lies in the absorption range of pollutants, should easily integrate with the reactor design into one reactor unit, and maximizes the irradiation of photocatalyst surfaces. Such a light system can be achieved by distributing light from LEDs using optical fibers. In optical fibers, light is incident on the surface of the photocatalysts. Illumination can be maximized by immobilizing the photocatalysts on the optical fibers [161,176,177]. This arrangement permits light to reach the photocatalytic nanoparticles more efficiently because less light is absorbed by other particles in the solution. In addition, immobilizing photocatalysts on optical fibers provides an innovative photocatalyst recovery approach which can mitigate membrane fouling by photocatalysts during recovery and poor irradiation of photocatalytic membranes surfaces. Nguyen et al. [176] developed an optical fiber-based reactor for the reduction of carbon dioxide with water to produce fuels under artificial light and natural sunlight. TiO_2_–SiO_2_ nanoparticles and magnetic TiO_2_–SiO_2_ nanoparticles were immobilized on separate optical fibers. The TiO_2_–SiO_2_ system achieved a 0.023% quantum yield, while the magnetic TiO_2_–SiO_2_ system had a 0.5% quantum yield. Consequently, the study showed that the addition of magnetic nanoparticles influenced product selectivity and visible light absorption. Zhu et al. [177] designed an optical fiber light system where the total light reflection was aided by an aluminum fixed jacket. The designed system achieved a trichloroethylene removal efficiency of 86.66%. Based on the design, it is evident that the use of an optical fiber system can overcome light scattering and save energy [178].

### 4.4. Geometry

Irradiation source distribution has a major influence on PMR geometry. The geometry must be tailored to maximize irradiation collection while lowering investment and energy expenses. Batch and continuous PMRs have been developed with annular and rectangular (flat plate) geometries (Figure 10). In comparison to annular PMRs, the flat plate shape is more appealing since it is scalable and provides an ideal setup for effective excitation of visible-light active photocatalysts [178]. Furthermore, the flat plate design allows for unrestricted fluid flow with no recirculating or dead zones, resulting in a plug flow reactor that can function at a variety of fluid flow rates. This is also advantageous for the flat plate shape, since the flow regime must be carefully chosen to avoid catalyst surface damage caused by high velocities, shear stress, and turbulence. Sannino et al. [179] designed a flat plate configuration reactor for the degradation of methyl blue operating under continuous mode. The design depended on the light source, fluid dynamics, and adsorption photocatalyst properties.

### 4.5. Incorporation of Magnetic Photocatalytic Nanoparticles

The use of magnetic photocatalytic nanoparticles has shown the potential to yield better designs of PMRs. The engineering of magnetic composites with cobalt is particularly promising and has had a tremendous influence on the morphology of magnetic nanoparticles and has also enhanced their magnetic and optical properties [183,184,185,186,187]. Spinel ferrite is one of the most frequently designed magnetic nanoparticles with photocatalytic and superparamagnetic properties [90]. These properties qualify spinel ferrite nanoparticles for application in the treatment of water and wastewater. The separation of spinel ferrite nanoparticles from water matrixes can be achieved by magnetism; this makes the recovery of photocatalysts easier and more effective, reducing the possibility of photocatalysts leaching into the environment. Kefeni et al. [186] developed α-Fe_2_O_3_ nanoparticles via coprecipitation for application in acid drainage treatment. The nanoparticles exhibited superparamagnetic characteristics with 5.6 emug^−1^; having a 5.6 emug^−1^ saturation magnetism shows that α-Fe_2_O_3_ nanoparticles can be recovered using an external magnetic field. Similarly, Masunga et al. [183] synthesized CuFe_2_O_4_ nanoparticles via coprecipitation. The CuFe_2_O_4_ nanoparticles had a bandgap of approximately 1.6 eV and 63.3 emug^−1^ saturation magnetization. The lower band gap indicated that the CuFe_2_O_4_ nanoparticles are potential candidates for photocatalysis applications.

In PMRs, magnetic photocatalytic nanoparticles can be immobilized either as part of the membrane or on the optical fibers (Figure 11). Even though wastewater has a high turbidity, immobilization of the nanoparticles on the optical fibers can minimize light attenuation and maximize light absorption by the nanoparticles, improving photocatalytic degradation performance.

In addition to superparamagnetic and photocatalytic properties, magnetic nanoparticles are nontoxic and have high stability properties. Cobalt magnetic nanoparticles have been integrated in PMRs; see, for example, the work of Shevale et al. [189]. The investigation obtained a complete degradation of Azorubin S colorant using TiO_2_-CoFe_2_O_4_ nanocomposites under visible light. The optical band gap was tuned from 3.2 eV of bare TiO_2_ to 2.8 eV of the magnetic nanocomposite, resulting in high photonic and quantum efficiency. Similarly, Nazarkovsky et al. [190] prepared and tested CoFe_2_O_4_/SiO_2_/TiO_2_ for the degradation of carbamazepine. Optimal results were achieved, with 100% degradation of carbamazepine, and the catalysts were successfully recovered from the reaction environment using an external magnetic field. The migration of Co^2+^ ions from octahedral to tetrahedral oxygen coordination surroundings, enhanced the magnetic properties of nanomaterials. Elsewhere, Lu et al. [191] demonstrated the feasibility of CoOFe_2_O_3_@rGO@TiO_2_ as a potential photocatalyst in the degradation of dimethylhydrazine from synthetic water. The overall degradation efficiency obtained was higher than 70%. This demonstrated that magnetic nanoparticles have better control of the photocatalytic process, owing to the delay in charge recombination and enhanced activity under visible light irradiation.

## 5. Applications of PMRs in the Removal of Emerging Contaminants

PMRs have been used for the removal of emerging contaminants (ECs) from wastewater. Such pollutants include but are not limited to pharmaceuticals, hormones, personal care products, per- and polyfluoroalkyl substances (PFASs), disinfection byproducts, flame retardants, plasticizers, and endocrine-disrupting compounds (EDCs). Although the removal of several emerging pollutants has been widely reported [78,95,187,190], the practical application of PMRs at pilot and industrial scales is still rare. The validation of mathematical modeling data against experimental data is required for the scaling-up of PMRs. The results of modeling will assist in the design and optimization of products [191]. The efficacy of a designed PMR is dependent on reaction kinetics and fluid dynamics, hence the review has focused on these two models [166]. Wang et al. [166] demonstrated the dependence of flat plate PMR efficacy on kinetics and fluid dynamics. The authors used computational fluid dynamics simulation and Langmuir–Hinshelwood reaction kinetics to simulate the photocatalytic degradation of dimethyl sulfide. The concentration of dimethyl sulfide kept decreasing along with fluid flow, and the pollutant concentration was high in the middle and low at the margins, demonstrating that light intensity was high at the edges and low in the center. The best dimethyl sulfide degradation efficiency was found to be 80%. The hydrodynamics of an annular PMR were modeled by Casado et al. [36] The inflow was not homogeneous throughout the reactor, and there was a dead zone at the outlet with a low velocity magnitude, according to the continuous fluid dynamics analysis. However, because of the underdeveloped reactor flow dynamics, formaldehyde degradation was poor. Similarly, Tisa et al. [33] developed a fluidized bed PMR based on kinetic and fluid dynamic simulations for the degradation of phenol. The magnitude of the velocity fluctuated across the reactor. For example, the velocity was higher towards the inlet but lower in the center zone due to the presence of pores on the goethite catalyst. With a Reynolds number of 300, the fluid flow was laminar, ensuring complete fluidization. Interestingly, the reaction kinetics revealed that the concentration of phenol declined across the length of the reactor and that the conversion increased with reaction time for reactor lengths greater than 10 m.

Considering large-scale PMR applications, initiatives have been made to scale up PMRs from lab to pilot size [13,40,119]. Plakas et al. [40] fully automated a continuous flow suspended PMR, with a processing capacity of 1.2 m^3^/day and a UV power of 52 W. The reactor achieved a diclofenac degradation rate of 99%. However, as previously stated, suspended photocatalysts have practical scaling constraints. Immobilized photocatalysts are a top research and development priority in PMR systems and they must be validated in practical pilot scale conditions for performance and stability under long-term continuous usage. The radiation of PMRs under large-scale operation is another source of concern [163]. Optical fibers coupled to UV LEDs have been used at pilot scale [13,119]. Athanasiou et al. [13] operated a 50 m^3^/day pilot plant using multichannel membranes and optical fibers powered by solar radiation. The optimal alignment of the optical fibers in relation to the membrane channel design is critical and is controlled by the PMR’s processing capacity.

To highlight the application of PMRs in the removal of ECs, the following section will focus on the removal of EDCs by PMRs. It is well-documented in the literature that EDCs and their metabolites are partially removed by standard WWTPs and can be quantified in water at concentrations of up to g L^−1^ [55,192,193,194]. Regardless of the concentration level, the presence of EDCs in water and continuous exposure raises several concerns due to their toxicological chronic consequences which remain unknown. Most of the applied treatment processes, including biological treatment and adsorption, do not completely remove EDCs from polluted water. Consequently, a lot of research has focused on developing effective technologies to remove EDCs from water as an alternative to the present treatment methods. PMRs have therefore been presented as a promising and viable solution to overcome existing limitations. Table 3 provides a summary of other studies that have been carried out for the removal of various contaminants from water. For the purpose of this review the subsequent section will focus solely on the application of PMRs in the removal of EDCs.

Augugliaro et al. [8] presented a solar PMR model that used a suspended catalyst linked to an NF membrane module for the application of PMRs in removing EDCs. Lincomycin was degraded in aqueous solutions using TiO_2_-P25 that had been exposed to sunlight. The study also investigated the influence of membrane type on lincomycin removal. The DK2540C membrane achieved 93.64% and 97.78% lincomycin removal, while the DL2540C membrane achieved 82.6% and 91.3% using 25 µM and 75 µM initial concentrations of lincomycin. It was further found that the degree of TOC accumulation depended on solar irradiance and initial lincomycin concentrations. The light source provided insufficient photons to radiate the reactive mixture. Elsewhere, Pastrana-Martinez et al. [42] investigated the photocatalytic degradation of diphenhydramine (DPH) under UV–vis irradiation using a TiO_2_/graphene oxide (GO)-immobilized PMR. In terms of DPH elimination, the inclusion of GO increased the efficiency from 43% to 73%. The bandgap value of TiO_2_ was reduced by GO, which increased the visible light responsiveness as a result. The authors further investigated the effect of Cl^−^ _(aq)_, and the data showed that DPH degradation efficiency was reduced. This was ascribed to the ability of Cl ions to act as holes and radical scavengers.

The removal of 4-nitro-phenol using immobilized TiO_2_ was carried out [84]. In the specific design of this PMR, the irradiation source was placed on the recirculation tank or in the cell containing the membrane. The study demonstrated that the rate of 4-nitro-phenol photodegradation was strongly affected by the UV irradiation mode, as the reactor with the immersed lamp was three times more efficient than that with the external lamp, although the power was four times higher.

In two separate studies, Luster et al. [209] and Horovitz et al. [48] both reported the degradation of carbamazepine using a N-doped TiO_2_-coated PMR. The performance of Horovitz’s model was improved by recirculating the treated water through the photocatalytic membrane, resulting in 90% pollutant removal. Luster et al. [209] further went on to investigate the effect of different Ca and Mg ions on photocatalytic oxidation. The results showed that the highest degradation rate of pollutants was observed at a neutral pH and that the presence of Ca^2+^ and Mg^2+^ in basic conditions had no noticeable effect on pollutant degradation. However, the presence of Ca^2+^ at a neutral pH resulted in about a 30% decrease in photocatalytic degradation. From the above, it could be concluded that the presence of salts and dissolved organic debris can have the following effects on system performance in PMRs: (a) decreased photocatalytic performance or (b) increased membrane fouling. In the same vein, Pastrana-Martinez et al. [210] studied the influence of dissolved NaCl on the performance of PMRs. The reactor was used to study the extent of the degradation and mineralization of methyl orange in the continuous mode under near UV–vis and visible light irradiation. Maximum deteriorations of 65% in near UV–vis light and 19% in visible light were achieved. The inclusion of NaCl (0.5 g L^−1^) resulted in a modest reduction in methyl orange decomposition, regardless of the membranes used, because Cl ions operated as holes and hydroxyl radical scavengers.

As highlighted in Section 1.2, immobilized PMRs can be operated either under dead-end flow mode or crossflow mode [49]. Moslehyani et al. [71] designed batch PMRs operating under dead-end mode for the removal of hydrocarbon from bilge water. After 2 h of operation, 99% of hydrocarbons were removed from the water. It is important to note that the absence of a recycling stream resulted in low photocatalytic efficiency due to inadequate contact between molecules, photocatalysts, and light. In another typical PMR operating in dead-end mode, a Ag-TiO_2_-coated alumina membrane was employed for the degradation of Rhodamine [38]. The development of mesoporous material from a type-IV adsorption isotherm was proposed by membrane porosimetry results and Rhodamine was degraded at a rate of 1.007 mg m^−2^h^−1^. Despite the encouraging findings, the authors underlined that dead-end operation resulted in the accumulation of separated substrates on the membrane surface and eventually the formation of a cake layer that reduced photocatalytic performance.

To improve the remediation of EDCs from water, Molinari et al. [211] developed another set of PMRs. The authors proposed two PMR configurations: pressurized and depressurized. Tamoxifen (TAM) and gemfibrozil (GEM) were degraded in both systems using suspended TiO_2_-P25 as a catalyst. A flat sheet NF membrane with a permeate flux of 38.6 Lm^−2^ h^−1^ and a pressure of 6 bar was utilized in the pressurized system. The researcher obtained a GEM degradation rate of 60%, with a TOC rejection rate of 62% (Figure 12), which showed that a membrane with a high rejection rate was required. TAM was able to pass through the membrane, and its photodegradation intermediates were found in the permeate. Additionally, flux decline was attributed to fouling and deposition of TiO_2_ on the membrane surface. On the other hand, submerged capillary polyethersulfone (PES) membranes running at a flow of 65.1 Lm^−2^ h^−1^ and a vacuum of 0.133 bar were used in the depressurized setup to control fouling. The results demonstrated that the membrane was not able to reject GEM and its oxidation products, as they were present in the permeate stream.

With fouling being the major drawback to PMRs, Sarasidis et al. [59] developed a membrane back-flushing strategy for fouling control. The degradation of diclofenac was tested using a TiO_2_-submerged PMR working in continuous mode. Subsequently, there was a 99% degradation rate and a 66% TOC mineralization rate. The partial degradation was attributed to the generation of recalcitrant byproducts. After 9 min of filtration, periodic back-flushing every 1 min effectively controlled membrane fouling, allowing for steady continuous operation. Doll et al. [72] employed a crossflow MF membrane and adopted a similar periodic back-flushing method for the photodegradation of carbamazepine. The split PMR, which included a UV light source, TiO_2_, and an Al_2_O_3_ ceramic membrane, achieved a degradation efficiency of 98% in just 24 h. In another study by Wang et al. [51], an integrative submerged PMR system was developed for carbamazepine degradation. A 1 L tubular borosilicate glass photoreactor, a C-N-S doped Titania photocatalyst, 240 visible light LED modules, and a hollow fiber MF membrane module were used in the PMR design. In just 10 h, this system was able to remove 69% of the carbamazepine. In a similar setup to that of Wang et al. [51], Chin et al. [212] employed four UV lamps to remove bisphenol A (BPA) instead of employing LEDs. BPA was 97% degraded in 90 min, and TOC was removed by 93% in 120 min thanks to photocatalytic mineralization and simultaneous filtering.

Elsewhere, Darawna et al. [213] investigated the correlation between feed matrix and membrane fouling. Ibuprofen and diclofenac were removed from tap water, ultrapure water, secondary effluent, and primary effluent by the PMR. The study proved that the feed matrix has a considerable impact on EDC removal performance, and the order of removal was ultrapure water > tap water > secondary effluent > primary effluent. Despite the marked levels of pollutant photodegradation (Ibuprofen, >70%; diclofenac, 100%), poor mineralization rates were recorded (14% and 23% for primary effluent and secondary effluent, respectively. Tap water and secondary effluent treatments had no effect on permeate flux; however, primary effluent remediation led to the development of a fouling film, which reduced permeate flux. Furthermore, Fernandez et al. [54] also employed a submerged PMR with air bubbling to remove 33 emerging pollutants, including sulfamethoxazole, ketoprofen, ibuprofen, diclofenac, carbamazepine, and others. The degradation efficiency reached up to 95%, and the results were closely related to the photodegradation kinetics of the model pollutants. All pollutants with kinetic constant (*k*) values greater than 0.0544 min^−1^ were efficiently eliminated in the PMR.

Another interesting approach for diclofenac degradation was also investigated by Fischer et al. [63]. The authors developed two immobilized TiO_2_ on the surface of PES and polyvinylidene fluoride (PVDF) hydrophilic membranes. TiO_2_/PES performed better compared to TiO_2_/PVDF, with 68% and 55% removal, respectively. Lower degradation rates (Figure 13) were due to higher diclofenac concentrations in water, resulting in an imbalance in catalyst reactive sites and diclofenac substrates; hence, recycling the permeate stream could have improved the quality of the permeate.

A slurry PMR was devised by Asha and Kumar [214] for the degradation of sulfamethoxazole. The design comprised a flat submerged membrane and granular activated carbon-TiO_2_ P25 (GAC-TiO_2_) photocatalysts. Complete sulfamethoxazole degradation was achieved by using a photocatalyst loading capacity of 529.3 mg L^−^^1^ and 125 min residence time. At greater doses, however, sulfamethoxazole degradation was reduced to less than 50%.

The presented examples of the photodegradation of EDCs by PMRs demonstrate that a thorough understanding of feedwater quality is required to build a PMR process that effectively removes EDCs. Molinari et al. [215] demonstrated a TiO_2_-P25 slurry PMR for furosemide and ranitidine degradation with continuous recirculation. Different membrane rejection percentages were achieved in the dark with ranitidine (10–60%) and furosemide (5–30%) (Figure 14), showing that the proposed membrane could not reject ranitidine and furosemide effectively.

Furthermore, the membrane was only able to keep the photocatalysts contained during the photodegradation studies. These data imply that choosing the right membrane is an important part of designing experiments with PMRs. The recent studies discussed above have demonstrated the benefits and the potential of PMRs in the removal of emerging pollutants and have pinpointed the necessity of understanding the factors that govern the performance of PMRs.

## 6. Conclusions and Prospects

PMRs are a potential technology for the remediation of recalcitrant pollutants such as EDCs. An ideal PMR should offer high throughput, be low-cost, and have energy efficiency; this can be achieved by selecting suitable materials for construction, using magnetic photocatalytic nanoparticles (which are responsive to visible light and have high photocatalytic activity), using optical fiber systems to mitigate light scattering and save energy, considering reactor geometry, and by designing a channel diameter that minimizes the risk of pressure drops during operation. Among PMR designs, immobilized PMRs offer a more suitable configuration compared to slurry reactors. Despite high mass transfer and photodegradation performance, slurry PMRs still suffer from high operating costs and photocatalyst-induced membrane fouling. This can be mitigated by immobilized PMRs, in which photocatalysts are immobilized on a membrane module; additionally, light scattering can be improved by immobilizing photocatalysts on optical fibres. However, the performance of PMRs has been limited by factors such as membrane rejection and configuration, photocatalyst loading capacity, light sources, and residence times. This leads us to place emphasis on the need for process optimizations (e.g., fluid flow and reaction kinetics). Further, the scaling up of PMRs is one of the challenging topics in photocatalytic reaction engineering due to the unavailability of simpler mathematical models preserving the fundamental aspects of rigorous models which can be easier to utilize for scale-up and design objectives. Should these constraints be overcome, the use of PMRs at the industrial level will be completely realized. This review has connected reaction kinetics and transport phenomena to come up with potential design equation that can produce an acceptable reactor design. Finally, the use of PMRs in wastewater has led not only to the degradation of EDCs but also to the remediation of oily water, dye, heavy metals, pesticides, and wastewater.

## Figures and Tables

**Figure 1 membranes-12-00745-f001:**
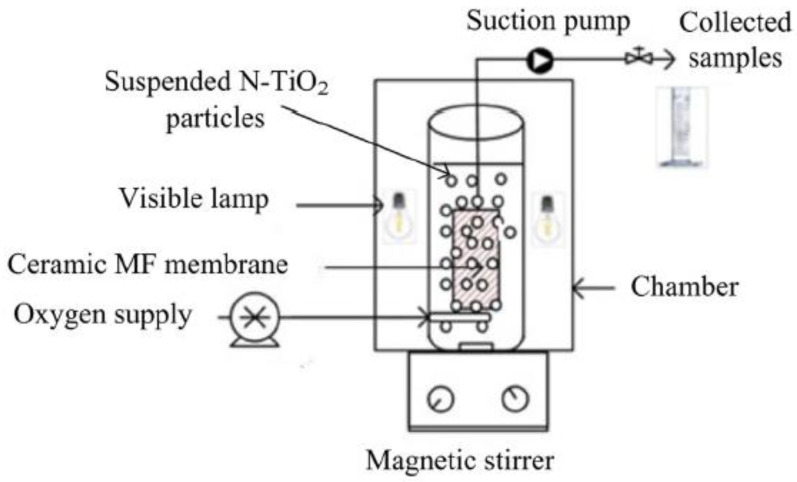
Diagram of submerged PMR with suspended N–TiO_2_. Reproduced with permission from [53], Copyright 2020 Elsevier.

**Figure 2 membranes-12-00745-f002:**
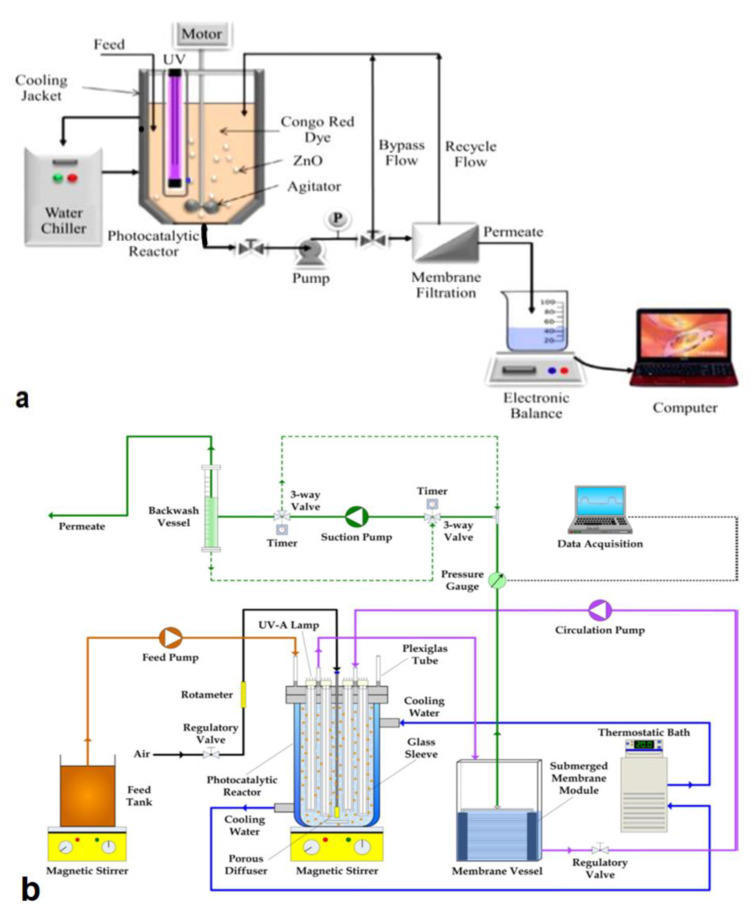
Schematic diagram of a typical split-type PMR (**a**) with suspended ZnO for degrading Congo red dye from wastewater, reproduced with permission from [11], Copyright 2014 Elsevier; and (**b**) a laboratory pilot system for the degradation of diclofenac in wastewater, reproduced with permission from [59], Copyright 2014 Elsevier.

**Figure 3 membranes-12-00745-f003:**
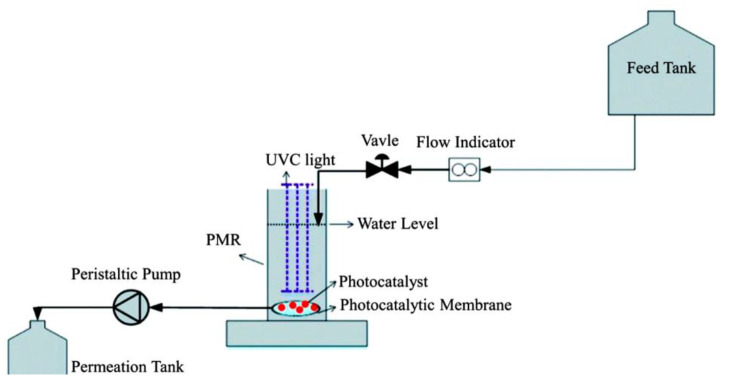
Schematic diagram of a dead-end PMR combining photocatalysis and membrane filtration. Reproduced with permission from [71], Copyright 2015 RSC Publishing.

**Figure 4 membranes-12-00745-f004:**
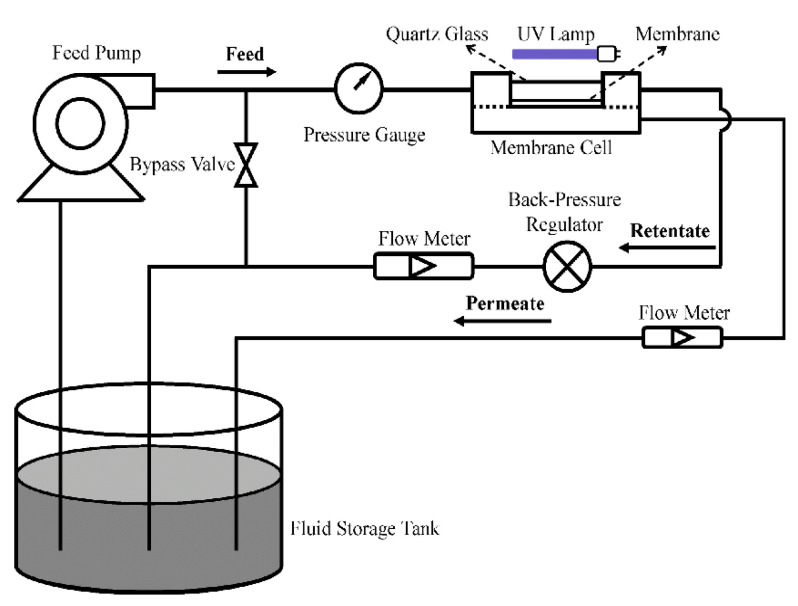
A representation of a PMR operating in crossflow mode. Reproduced with permission from [44], Copyright 2014 Elsevier.

**Figure 5 membranes-12-00745-f005:**
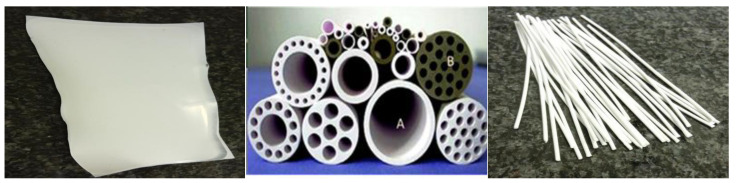
Different membrane configurations: flat sheet membrane was produced by authors of this study (**left**); hollow membranes (**center**). A—tubular hollow membrane; B—multichannel monolithic hollow membrane, reproduced with permission from [13], Copyright 2016 Elsevier. Hollow fiber membranes were internally produced (**right**).

**Figure 6 membranes-12-00745-f006:**
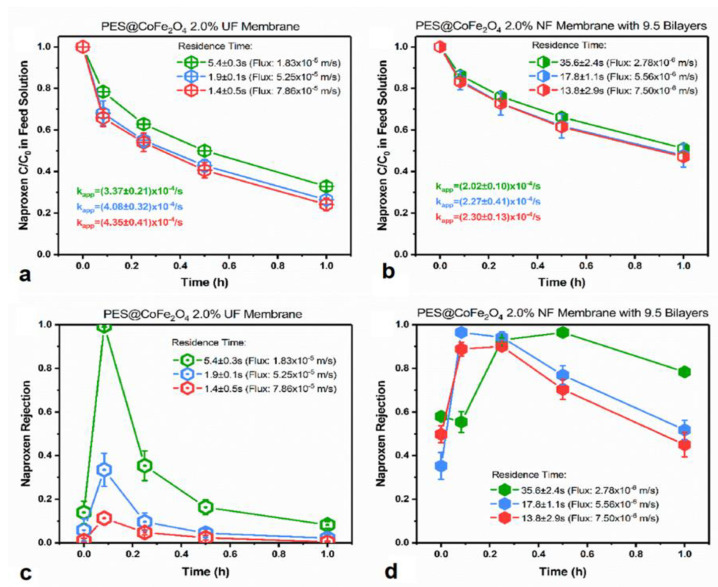
Effects of residence time on naproxen degradation and rejection: degradation by UF (**a**) and NF (**b**) membranes; rejection by the UF membrane (**c**) and the NF membrane (**d**). Reproduced with permission from [109], Copyright 2021 Elsevier.

**Figure 7 membranes-12-00745-f007:**
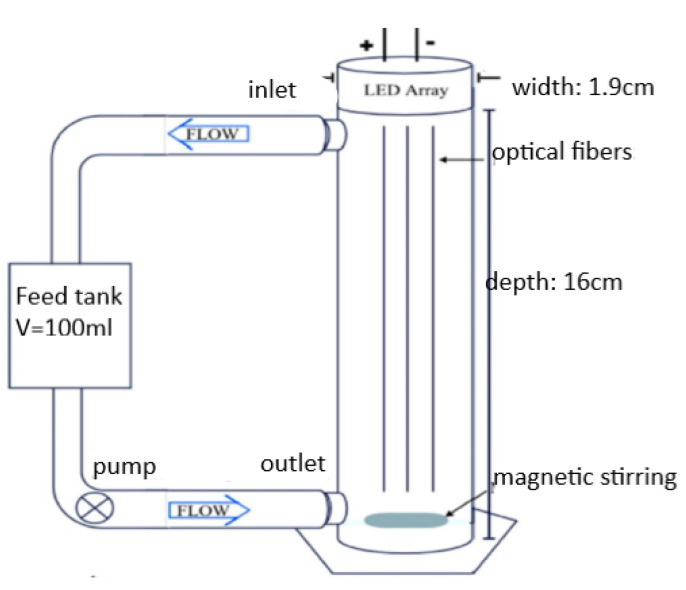
Schematic design of an optical fiber LED reactor developed by Tugaoen et al. Reproduced with permission from [119], Copyright 2018 Elsevier.

**Figure 8 membranes-12-00745-f008:**
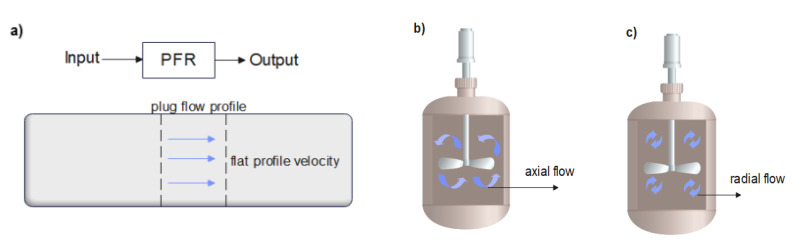
Fluid flow patterns: (**a**) plug flow (this figure is an adaptation of a figure contained in [150]); (**b**) axial flow; and (**c**) radial flow in reactors figures (**b**) and (**c**) are adapted from a figure contained in [149].

**Figure 9 membranes-12-00745-f009:**
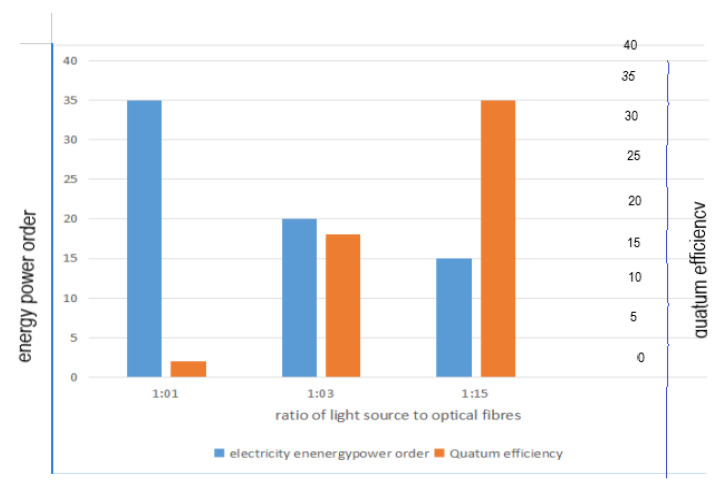
Relation between ω, quantum efficiency, and the number of optical fibers.

**Figure 10 membranes-12-00745-f010:**
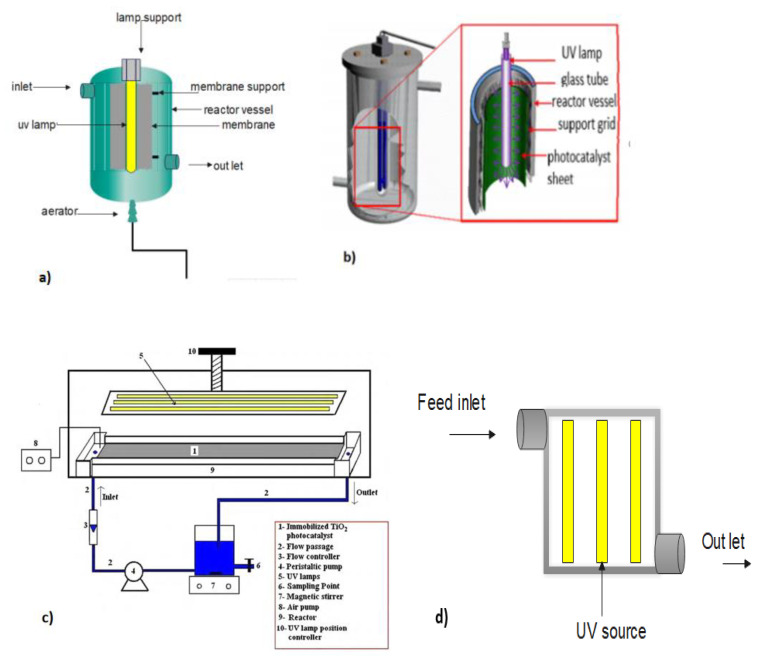
Illustration of different PMR geometries: (**a**) annular tubular reactor; (**b**) batch annular reactor, reproduced with permission from [180], Copyright 2019 Springer; (**c**) continuous flat rectangular reactor, reproduced with permission from [181], Copyright 2010 Elsevier; and (**d**) vertical flat plate reactor (this figure is an adaptation of figure contained in [182]).

**Figure 11 membranes-12-00745-f011:**
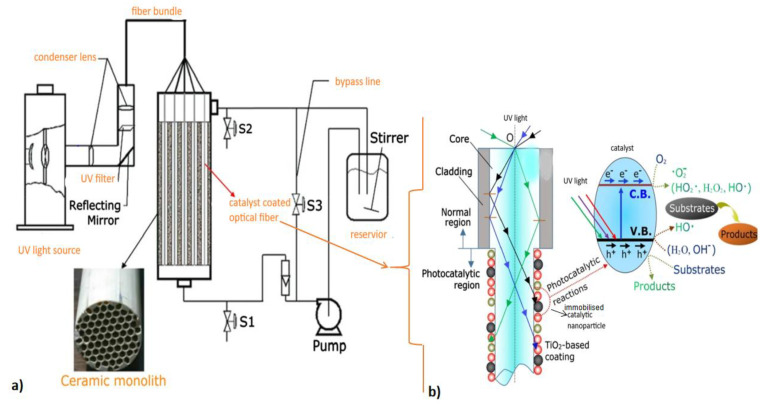
Ceramic PMR with optical fiber system (**a**), reproduced with permission from [174], Copyright 2005 Springer; and the mechanism of light transmission in photocatalytic optical fibers (**b**), reproduced with permission from [188], Copyright 2011 Springer.

**Figure 12 membranes-12-00745-f012:**
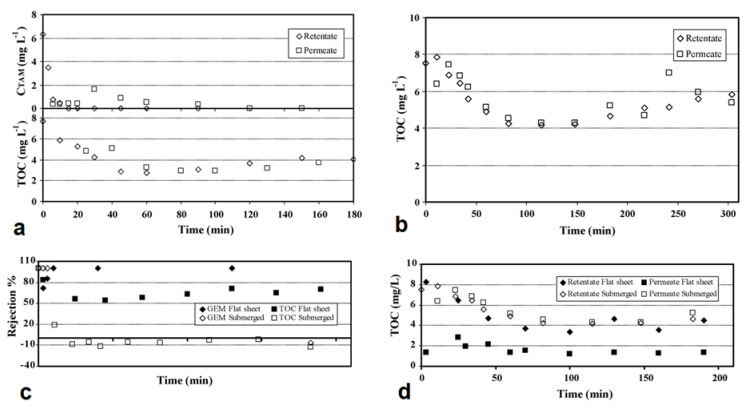
Degradation of drugs as a function of time: (**a**) degradation of citrate tamoxifen (TAM)—TAM concentration (CTAM) and TOC concentration; (**b**) Gemfibrozil (GEM) degradation; (**c**) GEM rejection in pressurized flat sheet membrane photoreactors; (**d**) GEM rejection in de-pressurized flat sheet membrane photoreactors. Reproduced with permission from [211], Copyright 2008 Elselvier.

**Figure 13 membranes-12-00745-f013:**
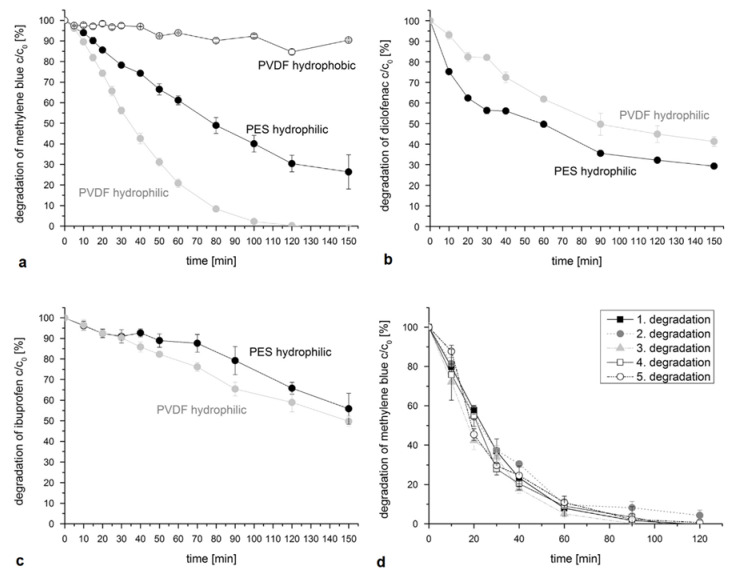
Pollutant photodegradation by hydrophilic PES membranes and hydrophobic PVDF membranes: (**a**) photodegradation of methylene blue; (**b**) photodegradation of diclofenac; (**c**) photodegradation of ibuprofen (**d**) photodegradation of methylene blue at different rates. Reproduced with permission from [63], Copyright 2015 Elselvier.

**Figure 14 membranes-12-00745-f014:**
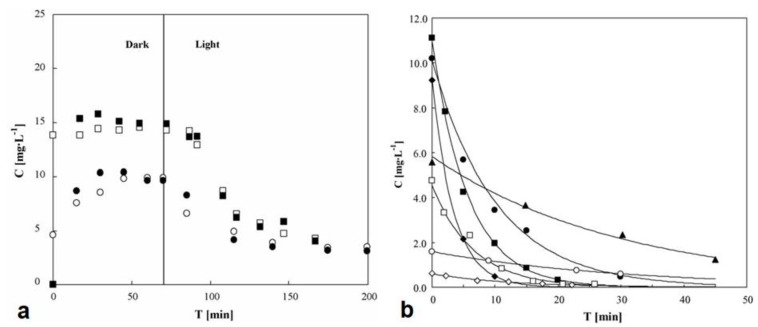
Concentration vs. irradiation time graphs for the removal of drugs: (**a**) substrate concentration under dark and light conditions at pH 3 (○ Furosemide retentate, ● Furosemide permeate, □ Ranitidine retentate, ■ Ranitidine permeate); (**b**) substrate concentration at TiO_2_ = 1 g L^−1^, CO_2_ concentration = 22 ppm (○ Furosemide, ● Ranitidine, ■ Phenazone, ▲ Clofibric acid, ◆ ofloxacin, □ carbamazepine, ◇ naproxen). Reproduced with permission from [215], Copyright 2006 Elselvier.

**Table 2 membranes-12-00745-t002:** Photocatalytic space–time yield (PSTY) vs. space–time yield (STY) (adapted from [50]).

	PSTY	High	Low
STY			
**High**		High illumination efficiency High mass ratio	Low illumination efficiency High mass ratio
**Low**		High illumination efficiency Low mass ratio	Low illumination efficiency Low mass ratio

**Table 3 membranes-12-00745-t003:** A summary of various applications of PMRs in water disinfection, heavy metal and wastewater reclamation, the removal of dyes, and oily wastewater treatment, as well as the treatment of pesticide wastewater.

	Reactor Type	Photocatalyst; Irradiation Source	Membrane Type	Water Matrix	Application	Highlights	Ref.
(1) Water disinfection	Immobilized PMR	TiO_2_ P25; UV light	Ceramic tubular membrane (0.8 μm pore size)	Simulated water containing bacteriophage P22	Virus removal	- Photocatalysis improved the LRV compared to simple UV disinfection - The proposed configuration is viable for working with turbid water, since radiation at the permeate side minimizes the scattering of light	[195]
	Submerged slurry PMR	TiO_2_ P25, UV	A flat sheet polyvinylidene fluoride (PVDF) membrane with a nominal pore size of 0.15 μm	Synthetic water	Virus removal	- The optimum operation was achieved with a 10 to 25 mg TiO_2_ load, at 40 Lm^−2^ h^−1^, and under intermittent suction mode -Fouling occurred at conditions above the optimum - 24 h continuous operation achieved; LRV of 99.99% was achieved	[196]
	Submerged slurry PMR	TiO_2_ P25, UV	Hollow fiber polyethylene membrane (0.4 μm pore size)	Municipal wastewater	Inactivation of bacteria	- Bacterial eradication was caused by membrane rejection - UV exposure, ROS oxidation, and adsorption at TiO_2_ surface successfully deactivated bacteria	[197]
	Immobilized PMR	TiO_2_, UVC lamp	Porous stainless steel MF membranes (0.2 μm and 0.5 μm pore sizes)	Synthetic wastewater	*Enterococcus faecalis, Escherichia coli,* and *Candida albicans* removal	- Immobilization of TiO_2_ on the membrane improved filtration performance and UVC attenuation	[198]
	Immobilized PMR	TiO_2_ solar UV–vis	N-doped TiO_2_-coated Al_2_O_3_ ceramic membrane	Natural surface water	Removal of MS2 *bacteriophage*	- LRV of 99.99% was achieved - Performance was affected by water quality - Pretreatment processes improved PMR performance, especially with high alkaline water and organic loading	[199]
(2) Treatment of heavy metals	Immobilized PMR	TiO_2_, nanozerovalent iron, UV light	Thin-film composite (TFC) membrane	Synthetic water	Reduction of Cr(VI)	- High water flow and antifouling capabilities were demonstrated by the membrane - Low Cr(VI) concentrations in permeate were achieved	[200]
	Immobilized PMR	TiO_2_/Ag NPs under visible light irradiation	Algae-decorated TiO_2_/Ag hybrid nanofiber membrane	Synthetic water	Photo-removal of Cr(VI)	- Algae inhibited electron and hole recombination, allowing electrons to effectively reduce Cr(VI) on the TiO2 surface - The PMR membrane continued to work effectively after 5 cycles, indicating that it could be useful for organic and heavy metals removal	[201]
(3) Treatment of reclaimed wastewater	Submerged PMR	TiO_2_, UV lamp	Tubular ceramic UF membranes	Municipal wastewater	Removal of secondary effluent organic matter	- Improved membrane fouling resistance with efficiency greater than 60% degradation - PMR efficiency was hampered by turbidity	[10]
	Immobilized PMR	ZrO_2_, UVC germicidal lamps	TiO_2_ tubular ceramic UF membranes	Municipal wastewater	Removal of secondary effluent organic matter	- 61% total organic carbon (TOC) removal was achieved after 5 h of operation - Optimum TiO_2_ of 1.5 g L^−1^ was used	[202]
	Slurry PMR	TiO_2_, UV lamp	Tubular ceramic membrane (0.1 μm pore size)	Municipal wastewater	Removal of secondary effluent organic matter	- During the first 60 min of PMR operation, permeate flux decreased - Organic chemical adsorption was pH-dependent	[203]
(4) Dye removal	Submerged PMR	ZnO or TiO_2_, UVC and UVA lamps	Flat sheet PES UF membrane	Raw textile and wood processing industry wastewaters	Removal of dye	- UVC lamps outperformed UVA lamps by a small margin - Initial wastewater concentration influenced colour removal considerably - Maximum degrading rate was found using an initial COD value of 150 mgO_2_ L^−1^	[204]
	Suspended PMR	ZnO, UV light	Poly piperazine amide NF membrane and polyamide UF membrane	Industrial dye wastewater	Removal of Congo red dye	- 65% Congo red removal - Minimum permeate flux (25%) was achieved using 0.3 g L^−1^ ZnO - NF membrane performed better in terms of turbidity reduction, colour removal, and rejection of total suspended solids	[58]
	Submerged PMR	TiO_2_ P25, microwave electrodeless lamps	PVDF hollow fiber membrane (0.2 μm)	Synthetic water	Reactive black 5 (RB5)	- 5 h of irradiation resulted in RB5 total decolorization and 80.1% TOC elimination - Increased photocatalyst loading from 0.5 to 6.0 g L^−1^ resulted in a 15.8% reduction in permeate flux	[205]
(5) Treatment of oily wastewater	Submerged PMR	TiO_2_, UV irradiation	PVDF hollow fiber membrane	Synthetic cutting oil wastewater	Removal of oil	- Photocatalytic degradation and water flux were negatively influenced by increasing feed concentration - Under optimal conditions, TOC degradation (80%) and oil rejection (90%) were achieved	[93]
	Immobilized PMR	TiO_2_,	Hollow fiber PVDF membrane	Oil recovery platform water	Degradation of surfactants	- Membrane performance was impaired by agglomeration of TiO_2_ NPs - 66.73% COD removal and 47.95 Lm^−2^h^−^1 membrane flux were achieved	[206]
(6) Removal of pesticides	Immobilized PMR	TiO_2_	Ceramic membrane	Synthetic water	Removal of diuron and chlorfenvinphos	- Diuron and chlorfenvinphos removals were 95% and 78%, respectively	[207]
	Slurry PMR	GO-TiO_2_, UV–vis		Synthetic water made from natural water and ultrapure water	Removal of diuron, isoproturon, atrazine, and alachlor	- In a natural water matrix, improved performance was more meaningful - Under visible light irradiation, TiO_2_ doped with GO demonstrated improved photocatalytic performance	[208]

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
