# Peer review of "Interdependence of Kinetics and Fluid Dynamics in the Design of Photocatalytic Membrane Reactors"

_membranes, 2022, doi:10.3390/membranes12080745_

Round 1
Reviewer 1 Report
This manuscript summarizes the factors that need to be considered for PMRs design. The topic is very interesting and falls within the scope of the journal. As PMR is a hot topic now, the review paper will significantly impact the reader. On the other hand, I could find several minor issues to improve the quality of the manuscript.
1. Section reorganization: The introduction section includes all the affecting factors, but I suggest separating the section as the introduction section and the affecting factors.
2. Table: I think the table summarizing current state-of-the-art research is very important in the review paper. The authors provided several examples as figures and text, but I highly recommend providing the table summarizing it. For example, section 1.1 can have a table summarizing the examples provided in sections 1.1.1, and 1.1.2.
3. Figure 5: In this section, the authors explained that the hollow fiber membrane has several advantages, but the images only showed flat sheet and tubular types. I recommend providing the picture images of the hollow fiber membrane as well.
4. Formatting issue: Line 322, 615
5. Font issue: Line 386-402
6. In the abstract, several models were proposed, such as the radiation emission and absorption model, but I cannot find the proper explanation for these models. It was briefly mentioned in section 1.3. As kinetic was one of the main issues in this manuscript, I ask the authors to provide more explanation about the kinetic models for PMRs.
7. The interdependence of kinetics and fluid dynamics was not well discussed, and the interdependence should be used in section 2, showing the environmental application.
Reviewer 2 Report
membranes-1760132
This paper provides a comprehensive review of the interdependence of kinetics and fluid dynamics in the design of photocatalytic membrane reactors. It is very well structured and written and fills the gap in this research field.
- The structure of the manuscript can still be improved. Normally Section 1 is for the introduction, followed by other main topics
- Section 1.1 is actually really broad. It needs to be broken down into a few sections. It will be much easier if a summary table is provided.
- Equations 6-8 are developed for the general reactor system. It would be better to include the actual kinetic reaction with a catalyst.
- One important aspect of PMR is membrane stability. The polymeric membrane can be degraded under prolonged UV exposure. It should be included in the discussion.
- Most figures are of poor quality (sometimes over-stretched). Please provide high-resolution figures.
- Lines 945-951: The issue of scaling up has long been acknowledged. It will be very interesting if any attempt to do so in recent work and lesion from those work will really benefit the future development of this research field.
Round 2
Reviewer 2 Report
All of the concerns have been addressed. I would recommend for publication.